# Population structure and antibiotic resistance of swine extraintestinal pathogenic *Escherichia coli* from China

Xudong Li [1,2,6], Huifeng Hu [1,2,5,6], Yongwei Zhu[1,3,4], Taiquan Wang[1,2], Youlan Lu[1,3], Xiangru Wang [1,3,4], Zhong Peng [1,3,4], Ming Sun [1], Huanchun Chen[1,3,4], Jinshui Zheng [1,2] ✉ & Chen Tan [1,3,4] ✉

Extraintestinal Pathogenic *Escherichia coli* (ExPEC) pose a significant threat to human and animal health. However, the diversity and antibiotic resistance of animal ExPEC, and their connection to human infections, remain largely unexplored. The study performs large-scale genome sequencing and antibiotic resistance testing of 499 swine-derived ExPEC isolates from China. Results show swine ExPEC are phylogenetically diverse, with over 80% belonging to phylogroups B1 and A. Importantly, 15 swine ExPEC isolates exhibit genetic relatedness to human-origin *E. coli* strains. Additionally, 49 strains harbor toxins typical of enteric *E. coli* pathotypes, implying hybrid pathotypes. Notably, 97% of the total strains are multidrug resistant, including resistance to critical human drugs like third- and fourth-generation cephalosporins. Correspondingly, genomic analysis unveils prevalent antibiotic resistance genes (ARGs), often associated with co-transfer mechanisms. Furthermore, analysis of 20 complete genomes illuminates the transmission pathways of ARGs within swine ExPEC and to human pathogens. For example, the transmission of plasmids co-harboring *fosA3*, *bla*$_{CTX-M-14}$, and *mcr-1* genes between swine ExPEC and human-origin *Salmonella enterica* is observed. These findings underscore the importance of monitoring and controlling ExPEC infections in animals, as they can serve as a reservoir of ARGs with the potential to affect human health or even be the origin of pathogens infecting humans.

Extraintestinal Pathogenic *Escherichia coli* (ExPEC) cause infections outside the gastrointestinal systems of humans and animals[1]. In humans, ExPEC are notorious for their contribution to urinary tract infection (UTI), bloodstream infection, neonatal meningitis, intra-abdominal infection, and soft-tissue infection[2]. About 30% of bacteremia with high mortality are caused by ExPEC[3]. They can also cause above 50% of the community-acquired UTIs among people with compromised immune systems, leading to substantial morbidity, mortality, and health care costs[3]. ExPEC strains colonize the intestinal tract but do not cause enteric disease, and differ from intestinal pathogenic *E. coli* which usually causes gastroenteritis or colitis in the host intestine[4]. ExPEC strains contain different virulence factors (VFs)

[1]National Key Laboratory of Agricultural Microbiology, Huazhong Agricultural University, Wuhan 430070, China. [2]Hubei Key Laboratory of Agricultural Bioinformatics, College of Informatics, Huazhong Agricultural University, Wuhan 430070, China. [3]Key Laboratory of Preventive Veterinary Medicine in Hubei Province, The Cooperative Innovation Center for Sustainable Pig Production, College of Veterinary Medicine, Huazhong Agricultural University, Wuhan 430070, China. [4]Frontiers Science Center for Animal Breeding and Sustainable Production, Wuhan 430070, China. [5]Present address: Centre for Microbiology and Environmental Systems Science, University of Vienna, Djerassiplatz 1, 1030 Vienna, Austria. [6]These authors contributed equally: Xudong Li, Huifeng Hu. ✉e-mail: jszheng@mail.hzau.edu.cn; tanchen@mail.hzau.edu.cn

from enteric pathogenic *E. coli*; however, these VFs are widely distributed beyond ExPEC and may contribute to host adaptation of commensal *E. coli* rather than toxicity of ExPEC[5,6].

To explore the diversity and transmission of ExPEC, traditional typing methods and large-scale genomic analysis were conducted. These provided evidence that more than half of the ExPEC strains from humans belonged to the phylogroup B2, which also included intestinal commensal *E. coli*[1,7–9]. The control of infections by ExPEC has been complicated by the emergence of ARGs, especially extended spectrum beta-lactamases (ESBLs) which can confer resistance to most beta-lactam antibiotics including three-/fourth-generation cephalosporins[10]. Genomic analysis has shown that most of the widespread ExPEC contained genes that determine resistance to multiple antibiotics[7,8]. Comparative genomics provided evidence of the crucial contribution of ARGs, especially the ESBLs, to the most wide-distributed ExPEC of ST131[11,12].

Animals, including chicken, duck, cattle, cow, swine and some domestic companion animals, are not exempt from the threat posed by ExPEC. For instance, avian pathogenic *E. coli* (APEC) is known to trigger colibacillosis in poultry, resulting in huge losses in poultry production[13–15]. The purpose of the One Health approach is to balance and optimize the health of people, animals, and ecosystems[16,17]. The research focused on ExPEC from animals mainly for three important reasons. One is to decipher the diversity and dynamics of ExPEC in animals to reducing the economic loss caused by these organisms[18,19]. The second is to reveal the relationship between ExPEC from animals and humans to provide more information about the dynamics of ExPEC to control these in humans[20,21]. Many studies have found that *E. coli* from animal reservoirs may be a potential source for human ExPEC, especially those from poultry[20–23]. Moreover, evidence of the pathogenicity of poultry-derived *E. coli* strains in causing extraintestinal infections in humans has been demonstrated through in vivo and in vitro experiments[24,25]. Notably, foodborne *E. coli* can also cause extraintestinal infections in humans, involving high-risk clones such as ST69, ST95, ST117, ST131, and ST648[26,27]. The third objective is to unravel the potential transmissions of ARGs from animal microbes to human pathogens, particularly those that can render commonly used and even so-called "last-resort" drugs ineffective[28,29]. Several plasmids and integrons are crucial in these transmission events[28–31]. Moreover, deciphering the transfer of ARGs between bacteria originating from both human and animal hosts is one of the primary missions of the One Health approach, which could provide important information for designing efficient strategies to mitigate resistance[32]. However, a comprehensive understanding of the contributions of animal ExPEC to human pathogens and their antibiotic resistances is lacking.

Previously, we and other groups found that ExPEC can lead to significant losses in the swine industry[18,19,33]. Swine ExPEC strains are recognized as significant disease agents, causing conditions such as UTI, meningitis, pneumonia, arthritis, and septicemia[34,35]. The population structure of swine ExPEC varied slightly across different datasets, primarily comprising the phylogroups A, B1, and D[18,33,36]. Additionally, uropathogenic *E. coli* (UPEC) from sows predominantly belonged to phylogroups B1, D, and E[35]. Furthermore, one study linked the density of industrial hog production to increased UTI rates in nearby human populations, suggesting that intense hog production may elevate UTI incidence in surrounding communities[37]. This observation was particularly concerning in light of the extensive use of antibiotics in the poultry and livestock industries, as antibiotic resistance has been found to be widespread in strains of *E. coli* from animal sources[38,39]. Despite these insights, a comprehensive understanding of the diversity and resistance determinants of swine ExPEC, as well as their relationships and potential spread to human pathogens, remains poorly documented.

In this work, we conduct large-scale genome sequencing and antibiotic resistance tests on 499 ExPEC isolated from disease samples of swine between the years 2011 and 2017 in several provinces of China. We explore the population structure of ExPEC, the relationship between drug resistance phenotype and genotype, and the putative contribution of these ExPEC to human infections.

## Results

### Swine ExPEC from China exhibit a highly diverse population structure

The ExPEC were isolated from diseased tissues of pigs from different piggeries in China, most of which were located in Central China. To ensure that each strain originated from distinct pigs, we exclusively isolated and preserved one representative ExPEC strain from each pig farm at a given time. In this study, we randomly selected 50-90 isolates for each year from 2011 to 2017 (Supplementary Fig. 1a), resulting in a total collection of 499 *E. coli* strains from 23 Chinese provinces, mainly from Hubei, Henan and Hunan provinces (Supplementary Data 1). These isolates were obtained from various organs of pigs, with 79.4% originating from the lung (Supplementary Fig. 1b). Each genome of the studied isolates was sequenced, producing approximately 1 Gbp of reads, which were subsequently assembled into contigs for further analysis.

We first performed comparative genomic analysis to gain insight into the diversity and population structure of ExPEC from swine in China. Phylogenetic analysis revealed that over 50% of the isolates belonged to group B1 (54%), with 30% originating from group A, 6.2% from group F, and 5.6% from the recently defined group G (Fig. 1 and Supplementary Data 1). Very few *E. coli* isolates studied here resided in phylogroups B2, D and E, with proportion for each less than 3%. Based on in silico multilocus sequence typing (MLST), 490 of 499 *E. coli* could be assigned to 75 sequence types (STs), with 9 strains representing novel STs. The top 5 STs with the most isolates (>5%) including ST410 (14.8%), ST88 (9.2%), ST101 (8.2%), ST117 (5.6%) and ST156 (5.2%), accounting for above 40% of the entire collection (Supplementary Fig. 2a). Among these STs, ST410, ST88, ST101, and ST156 belonged to phylogroup B1, while ST117 was categorized in phylogroup G. The remaining *E. coli* exhibited a high degree of diversity, with 60 STs, each comprising fewer than 10 members. We also inspected the isolated organs and times of all the *E. coli* on the basis of phylogenetic analysis. Our findings revealed that strains from a special organ or time point were distributed throughout the whole phylogenetic tree, suggesting an absence of association between the source or time of isolation and the phylogroup or ST. We further assessed the diversity of this *E. coli* collection by serotype prediction. The high level of diversity was confirmed by the results about H- and O- serotype combinations. Two hundred serotypes were found (Supplementary Data 1 and Supplementary Fig. 2b). The most common serotype was O89:H9, which contained 31 strains accounting for 6.21% of the total collection. The remaining serotypes included less than 20 isolates, and 177 serotypes included less than 5 isolates.

Additionally, to understand the epidemiology of these swine ExPEC strains, we compared them with all *E. coli* strains present in the EnteroBase database using core genome multi-locus sequence typing (cgMLST). As a result, we identified 39 cases of genetic relatedness between swine ExPEC and isolates collected from different sources, with a maximum separation of 10 alleles (Supplementary Data 2). Among these occurrences, 21 instances involved *E. coli* strains from the Enterobase database, with explicitly defined host types, including human, swine, and bovine. Human-host instances were the most frequently observed, with a total of 15 swine ExPEC isolates showing genetic relatedness to *E. coli* from various human body sites, such as blood and rectum. Among these, two strains belonging to ST648 (A71 and A91) and one human blood infection strain (EnteroBase ID, ESC_LA6470AA) differed by only 2 alleles, indicating very closely related genomes. Moreover, swine-derived ExPEC belonging to ST410, ST101, ST131, ST167, or ST744 were also observed to share a common

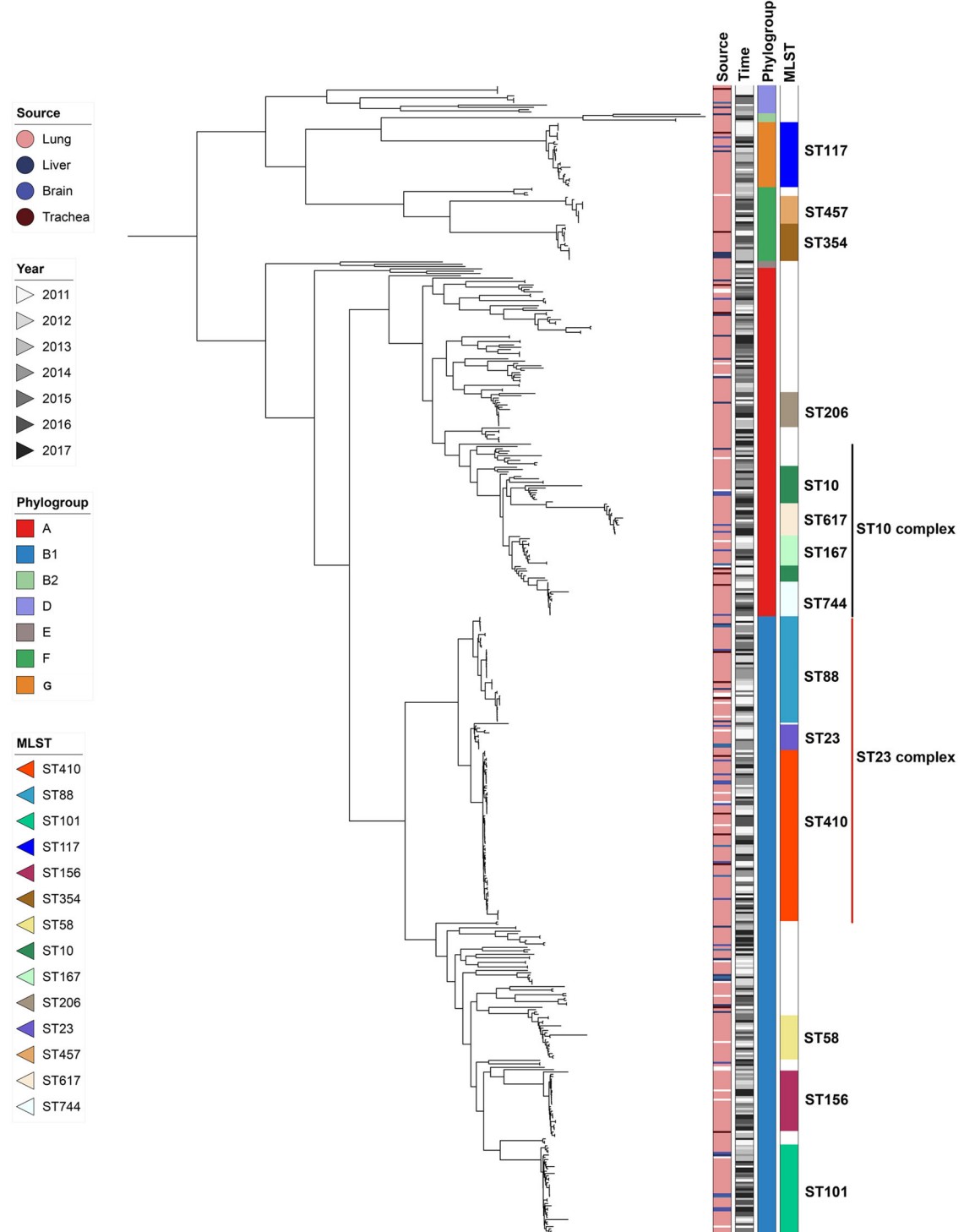

**Fig. 1 | Maximum likelihood phylogenetic tree of 499 swine-derived ExPEC isolates.** From left to right, the four columns display the primary tissue sources, the isolation time, the phylogroups, and prevalent STs of the isolates. Source data are provided as a Source Data file.

clonal origin with human-derived *E. coli*. This result underscores evidence of the zoonotic potential of ExPEC strains originating from swine.

### Genes encoding virulence factors associated with extra-intestinal infections are prevalent in swine ExPEC

Virulence factors (VF) play crucial roles in the interaction between *E. coli* and their vertebrate hosts. They have been used to define the pathogenicity features of *E. coli*, including those of different pathovars

of intestinal pathogenic *E. coli* (IPEC) and ExPEC[40]. To infer the relationship between swine ExPEC and enteric *E. coli*, the characteristic toxins of enteric *E. coli* pathotypes were predicted. We found that toxin genes determining enteric diseases were rarely present in these ExPEC strains (Supplementary Fig. 3). Specifically, 49 strains harbored one or more of the following: the Shiga-toxin (STX), heat-labile (LT) enterotoxin, heat-stable (ST) enterotoxin, intimin, and fimbrial adhesins (F4, F5, F18, and F41). These strains mainly belonged to phylogroups F, A, and B1. It is noteworthy that these strains may represent hybrid

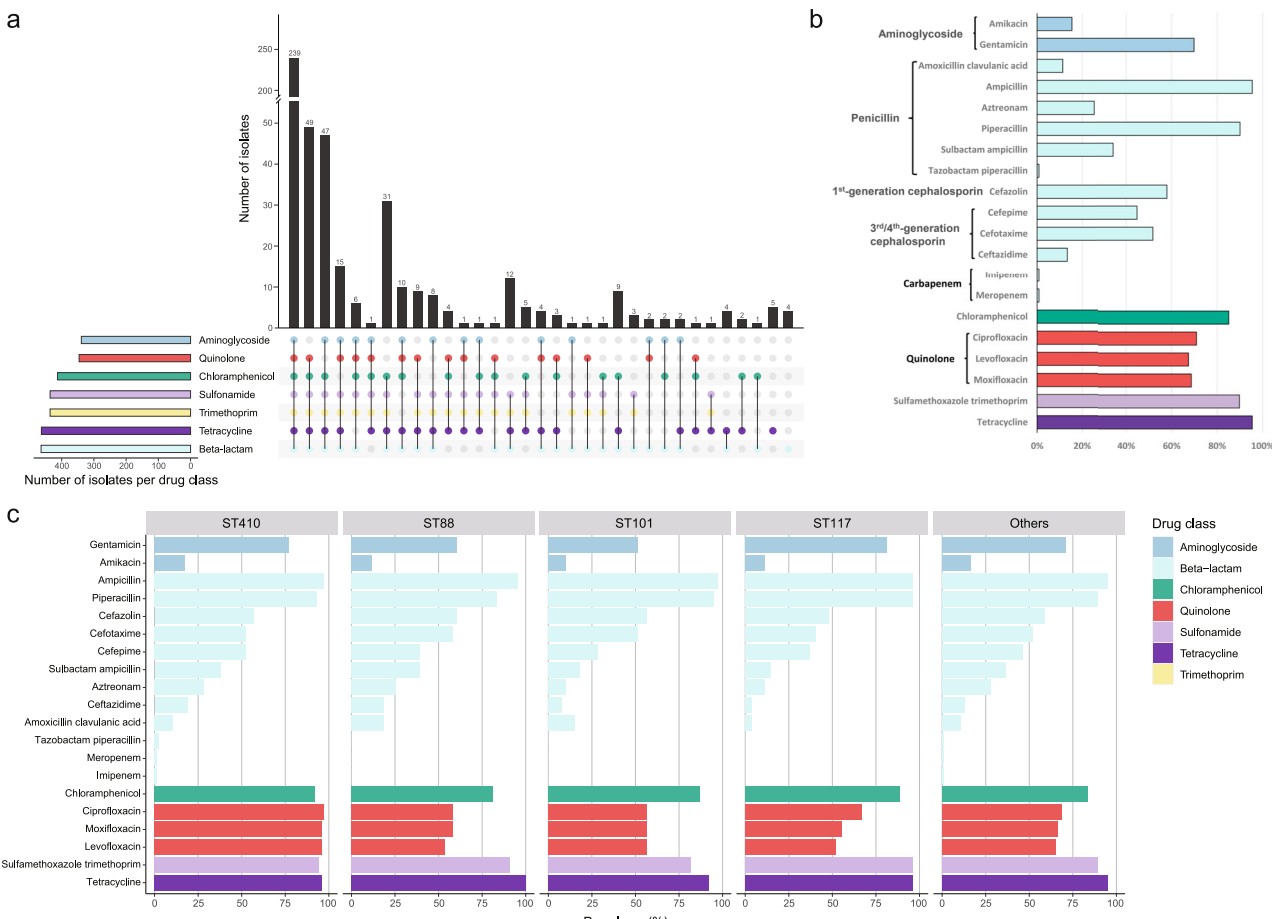

**Fig. 2 | Prevalence of antibiotic resistance phenotypes in 485 isolates. a** UpSet plot illustrating the diversity of antibiotic resistance profiles for ExPEC isolates. In the combination matrix at the base of the main bar chart, each column represents a distinct phenotypic profile. A colored dot within a column signifies resistance to at least one antimicrobial within a drug class, with the color corresponding to the respective drug class. The vertical bar chart displays the total number of isolates with specific phenotypic profiles, while the horizontal bar chart depicts the number of isolates resistant to each drug class. Sulfamethoxazole trimethoprim is a combination antibiotic, classified here as encompassing both sulfonamide and trimethoprim. **b** Histogram illustrating the prevalence of resistance to each test antimicrobial grouped by drug class in ExPEC strains. **c** The prevalence of antimicrobial resistance in four common STs of ExPEC. Source data are provided as a Source Data file.

pathotypes. Therefore, based on these virulence factors, we classified these 49 strains as four hybrid pathotypes, including ExPEC/ETEC, ExPEC/EPEC, ExPEC/EHEC, and ExPEC/EHEC/ETEC (Supplementary Data 3). The most prevalent hybrid pathotype is ExPEC/ETEC, comprising 39 strains in total. Roughly 70% of these strains contain the F5 fimbriae, all of which belonged to ST354/ST457.

We subsequently surveyed the five categories of VFs commonly found in human ExPEC[41] within the studied *E. coli* genome sequences. Regarding adhesins, genes associated with type 1 fimbriae (*fim*) and curli fibers (*csg*) were present in approximately 90% of the strains (Supplementary Data 3 and Supplementary Fig. 3). P fimbriae (*papC*) and antigen 43 (*agn43*) were identified in 10% and 18% of the strains, respectively, while other genes *afa* and *foc* were found in less than 5% of the strains. When considering iron uptake, the iron transport gene (*sitA*), aerobactin siderophore gene (*iutA*), and salmochelin siderophore gene (*iroN*) and were present in 69%, 66%, and 51% of the strains, respectively. While genes involved in yersiniabactin biosynthetic (*irp2*), heme uptake (*chuA*), and iron regulation (*ireA*) were detected in 36%, 16%, and 10% of the strains, respectively. When it comes to serum resistance, genes involved in protecting bacteria from complement-mediated lysis (*traT*), increased serum survival (*iss*) and colicin V synthesis (*cvaC*) were identified in 74%, 69%, and 32% of the genomes, respectively. Additionally, other genes within this category,

namely *ompA* and *kpsM*, were detected in 11% of the strains. As for the invasins, gene *ibeA* was found in only 4% of the strains. In terms of toxin genes, those associated with vacuolating autotransporter toxin (*vat*) and serine protease (*pic*) were present in 7% and 6% of the genomes, respectively. Other toxin genes such as *hly* (alpha-hemolysin), *cdt* (cytolethal distending toxin), and *cnf1* (cytotoxic necrotizing factor 1) were detected in less than 3% of the genomes.

## About 97% of swine ExPEC are classified as multidrug-resistant (MDR)

In the past decades, antibiotics were heavily used in poultry and livestock industries in China[42]. It has been reported that *E. coli* from food animals developed extreme resistance to frequently used antibiotics[29,43,44]. To assess the antibiotic resistance profiles of this *E. coli* collection, we conducted susceptibility test of these isolates to 20 antibiotics from 7 different classes, and found that all the isolates showed resistance to at least one of the used antibiotics (Supplementary Data 4). Among the 485 isolates with dependable antibiotic susceptibility outcomes (each test was validated using control strains, and only those passing quality control were considered reliable), MDR (multidrug-resistant, resistance to antibiotics from at least three classes[45]) strains accounted for up to 97% of the isolates (Fig. 2a). In addition, 239 of the 485 (49.3%) MDR strains were resistant to agents

spanning all 7 classes, with strains A376 and A430 exhibiting resistance to 18 antibiotics tested. Notably, among these MDR strains, 46 were hybrid pathotypes, of which 54.3% displayed resistance to agents across all 7 classes (Supplementary Data 4).

When the resistance to each antibiotic was analyzed, we found that most isolates were resistant to tetracycline (95.5%), ampicillin (95.5%), piperacillin (90.1%), sulfamethoxazole trimethoprim (89.9%) and chloramphenicol (85.2%) but only 0.8% of isolates were resistant to meropenem, imipenem, and tazobactam piperacillin (Fig. 2b). For aminoglycosides, about 70% of isolates were resistant to gentamicin and about 15.5% of isolates were resistant to amikacin. More than 65% E. coli showed resistance to each of the three fluoroquinolones, levofloxacin, moxifloxacin and ciprofloxacin. Additionally, resistance was observed in 10% to 60% of isolates for each of the other seven antibiotics within the beta-lactam class. Specifically, 51.8%, 13.4%, and 44.7% exhibited resistance to cefotaxime, ceftazidime, and cefepime, respectively, representing the third and fourth generation cephalosporins.

To assess the commonalities and specificities of antibiotic resistance among strains from different lineages, we analyzed the resistance profiles of strains belonging to the four most prevalent STs (Fig. 2c). We observed similar patterns in the prevalence of resistance to various drugs within each ST. Nonetheless, notable differences were observed, particularly in the prevalence of resistance to three fluoroquinolones and cefepime, which were significantly higher in ST410 compared to other STs. Furthermore, among the four top prevalent STs, only strains belonging to ST410 exhibited resistance to three antibiotics, including tazobactam piperacillin, meropenem, and imipenem.

## Swine ExPEC genomes harbor diverse and abundant antibiotic resistance genes

To understand the genetic determinants of antibiotic resistance, we searched all the sequenced genomes for genes and point mutations contributing to antibiotic resistance. We predicted a total of 96 ARGs and point mutations of 3 chromosomal genes (gyrA, parC and parE), which could contribute to resistance to antibiotics belonging to 12 classes (Fig. 3a and Supplementary Data 5). We found that all genomes harbored at least one of these ARGs. Each of the 13 ARGs that determines the resistance to aminoglycosides was contained by more than 5% isolates, among which strA and strB were present in 68% strains. The most frequently occurring sulfonamide resistance gene was sul2, encoded by 85% isolates, which is the highest prevalence among ARGs. Among the 4 phenicol genes, floR had a prevalence of 73%, with others less than 35%. One of the three types of tetracycline resistance protein genes, tet(A), was present in more than 70% of the strains, while both tet(B) and tet(M) were found in less than 35% of the isolates. When genetic determinants contributing to the resistance against fluoroquinolones were considered, we observed that, in addition to the high prevalence of the chromosomal gene mutations, three ARGs were identified, with oqxA and oqxB as the most prevalent one (approximately 45%). Besides, we also found the genes conferring resistance to colistin (mcr-1) and fosfomycin (fosA3) in 23% and 17%, respectively, of the tested isolates. Among the genes encoding β-lactamase, the most prevalent one was $bla_{TEM-1B}$ which was contained by more than 67% isolates. Other beta-lactamase encoding genes had prevalence above 5% include $bla_{CTX-M-14}$, $bla_{CTX-M-55}$, $bla_{OXA-1}$, $bla_{CMY-2}$, $bla_{CTX-M-65}$, $bla_{TEM-1A}$, and $bla_{TEM-141}$ most of which belong to extended-spectrum β-lactamase (ESBL) genes leading to the resistance to the three-/fourth-generation cephalosporins. When examining all ESBL genes present in our dataset, we found that as many as 262 strains harbored at least one ESBL gene. Moreover, among these, 25 strains were identified as hybrid pathotypes, specifically ExPEC/ETEC and ExPEC/EPEC (Supplementary Data 6). Additionally, 13 out of 15 strains with zoonotic potential also contained ESBL genes (Supplementary Data 2).

We conducted an additional analysis comparing the prevalence of ARGs of isolates from different STs (Supplementary Fig. 4). We found that the most prevalent ARGs within each class were similar, which is consistent with those for all the isolates. However, there were some over-presented ARGs in each of the four top prevalent STs. In ST410, we detected higher prevalence of ARGs including aph(3')-Ia, aac(6')-Ib-cr, rmtB, $bla_{CTX-M-55}$, $bla_{OXA-1}$, $bla_{TEM-141}$, catA1, catB3, oqxA, oqxB, qepA1, and tet(B). In ST88, the prevalence of resistance genes such as aac(3)-IIa, ant(3")-Ia, $bla_{CTX-M-65}$, qnsS1, and dfrA1 was higher compared to other STs. Additionally, mcr-1 and dfrA14 were more prevalent in ST101, and $bla_{CTX-M-15}$ and sul2 were common in ST117. Here, we further combined the comparative results of antibiotic resistance phenotypes among the four prevalent STs, especially noting that strains belonging to ST410 exhibited the highest resistance to three fluoroquinolones compared to the other STs. Therefore, we analyzed the distribution of genetic determinants responsible for fluoroquinolone resistance, including ARGs and chromosomal point mutations, within the ExPEC population (Supplementary Fig. 5). We observed mutations in three chromosomal genes (gyrA, parC, and parE) present in all ST410 strains, and the prevalence of fluoroquinolone resistance genes in ST410 was higher compared to the other prevalent STs. This could be one of the important genetic factors contributing to the prevalence of ST410. It's worth noting that all ST410 strains were identified as MDR. Furthermore, a similar pattern was observed in ST354, which presented the hybrid pathotype ExPEC/ETEC, with most strains belonging to this ST type containing ESBL genes.

## Highly diverse antibiotic resistance gene co-occurrences contribute to the high prevalence of multidrug-resistant (MDR) ExPEC

To investigate the genetic determinants behind the MDR and XDR (extensively drug-resistant, nonsusceptibility to at least one agent in all but two or fewer antimicrobial categories) E. coli, we examined the compositional profiles of ARGs in each genome. Among the 492 genomes containing at least 2 ARGs, we identified 406 combinations of distinct ARGs (Supplementary Data 7). More than 96% of the combinations were contained by only one or two genomes, with 340 genomes each harboring a unique set of ARGs. Each combination contained an average of 13 ARGs, with the largest one containing up to 28. More than two-thirds of the 406 combinations presented by more than 70% of genomes have over 10 ARGs, including 31 combinations found in 36 genomes containing more than 20 different genes. When drug classes targeted by these ARGs were predicted, we found that the putative MDR isolates accounted for up to 98% of the total, which is consistent with the test results described above. Our findings unveiled a striking diversity in the combinations of ARGs within the majority of the swine ExPEC genomes.

To investigate the simultaneous transfer of multiple ARGs, we conducted co-occurrence network analysis on the entire set of ARGs (Supplementary Data 8). The most prevalent sub-network consisted of sul2, floR, strA, and strB, forming the central core of the entire network, with most of the other ARGs having connections to this sub-network (Fig. 3b). This sub-network was identified in more than 200 of the 499 genomes and had the potential to give rise to MDR E. coli strains resistant to antibiotics from aminoglycoside, chloramphenicol and sulfonamide drug classes. The other highly clustered ARGs included tet(A), $bla_{TEM-1B}$, oqxA, and oqxB, conferring resistance to tetracyclines, beta-lactams, and fluoroquinolones, respectively. It is crucial to underscore the concurrent existence of ARGs targeting human drugs, such as $bla_{CTX-M-55}$, $bla_{CTX-M-14}$, fosA3, and mcr-1, in conjunction with ARGs targeting veterinary drugs. This issue carries notable significance, as the potential dispersal of veterinary drug-related ARGs, stemming from their misuse, indirectly amplifies the proliferation of co-existing ARGs against human drugs. Additionally, we observed that

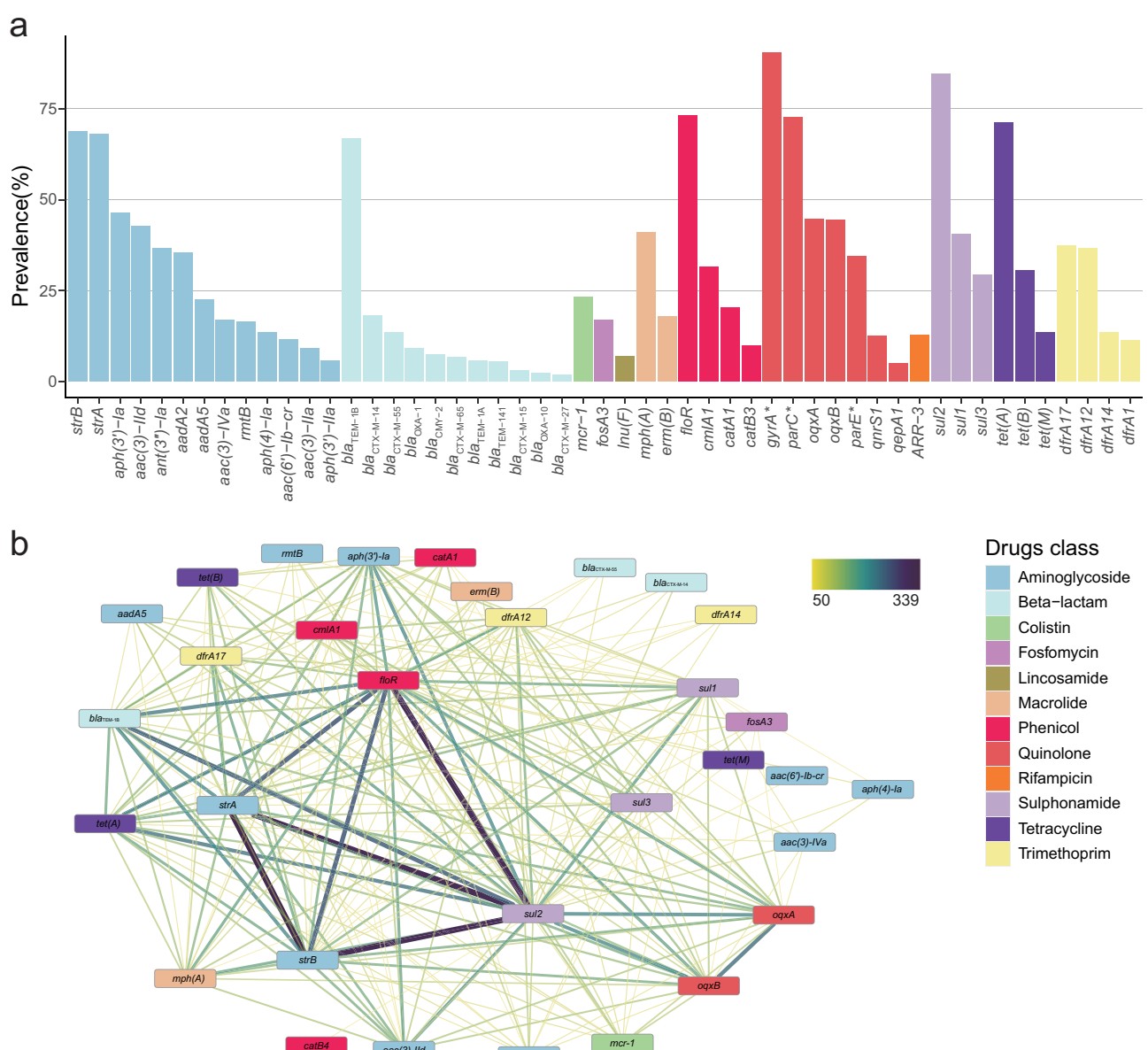

**Fig. 3 | Diversity of genetic determinants of antibiotic resistance in 499 ExPEC strains. a** Bar plot showing the percentage of genomes detecting each ARG or point mutation. Genes that contain point mutations leading to antibiotic resistance are marked with an asterisk. The color represents the class of antimicrobials to which genetic determinants confer resistance. The image displays only representative high-prevalence ARGs; you can find all the information in Supplementary Data 5. **b** Co-occurrence network of ARGs among ExPEC strains. The network was constructed based on the co-occurrence frequency of two-by-two ARGs in 499 genomes. The thickness and color depth of each link (edge) between ARGs (nodes) correspond to how frequently the two genes appeared together within the same genome. The image only visualizes genes that co-occurred together in at least 50 genomes. All connection information is available in Supplementary Data 8. Source data are provided as a Source Data file.

despite *mcr-3* being detected in only five genomes (*mcr-3.1* in four genomes and *mcr-3.5* in one genome), it is noteworthy that two of these genomes concurrently carry both *mcr-1* and *mcr-3* (Supplementary Data 8).

**Both chromosomes and plasmids are vectors of antibiotic resistance genes**

To decipher the vectors of ARGs, we chose 20 isolates from all the defined phylogenetic groups to obtain closed genomes. We obtained 62 closed replicons, 20 chromosomes and 42 plasmids, along with one incomplete plasmid sequence (Supplementary Data 9). The plasmids harbor one to four plasmid replicons, including IncFIB ($n = 17$), IncFIC ($n = 14$), IncI-gamma/K1 ($n = 10$), IncFIA ($n = 8$), and rep_cluster_2244 ($n = 8$). The average number of the ARGs contained by each

chromosome was 2.5, adding up to 50 ARGs across all 20 chromosomes. The tetracycline-associated resistance gene *tet(B)* was exclusively identified in chromosomes, not in plasmids. Another ARG that exhibited over-representation on chromosomes is *bla*CMY-2 (Supplementary Data 10). Of the 43 plasmids, thirty-seven harbored at least one ARG, with an average count of 6.2 and the highest observed being 15. AGRs over-presented in plasmid genomes included *aac(3)-IId*, *aadA2*, *mph(A)*, *dfrA12*, *oqxA*, *oqxB*, and *aph(3')-Ia*. The top prevalent ARGs were all frequently found in both chromosome and plasmid genomes, such as *sul1*, *sul2*, *floR*, *tet(A)*, *bla*TEM-1B, *strA* and *strB*, suggesting potential transfer between plasmids and chromosomes. In particular, we observed that certain ARGs have two copies located on distinct plasmids in the same isolate (Supplementary Data 9). For instance, in strain A430, each of the genes *aac(3)-IId*, *aadA2*, *bla*TEM-1B,

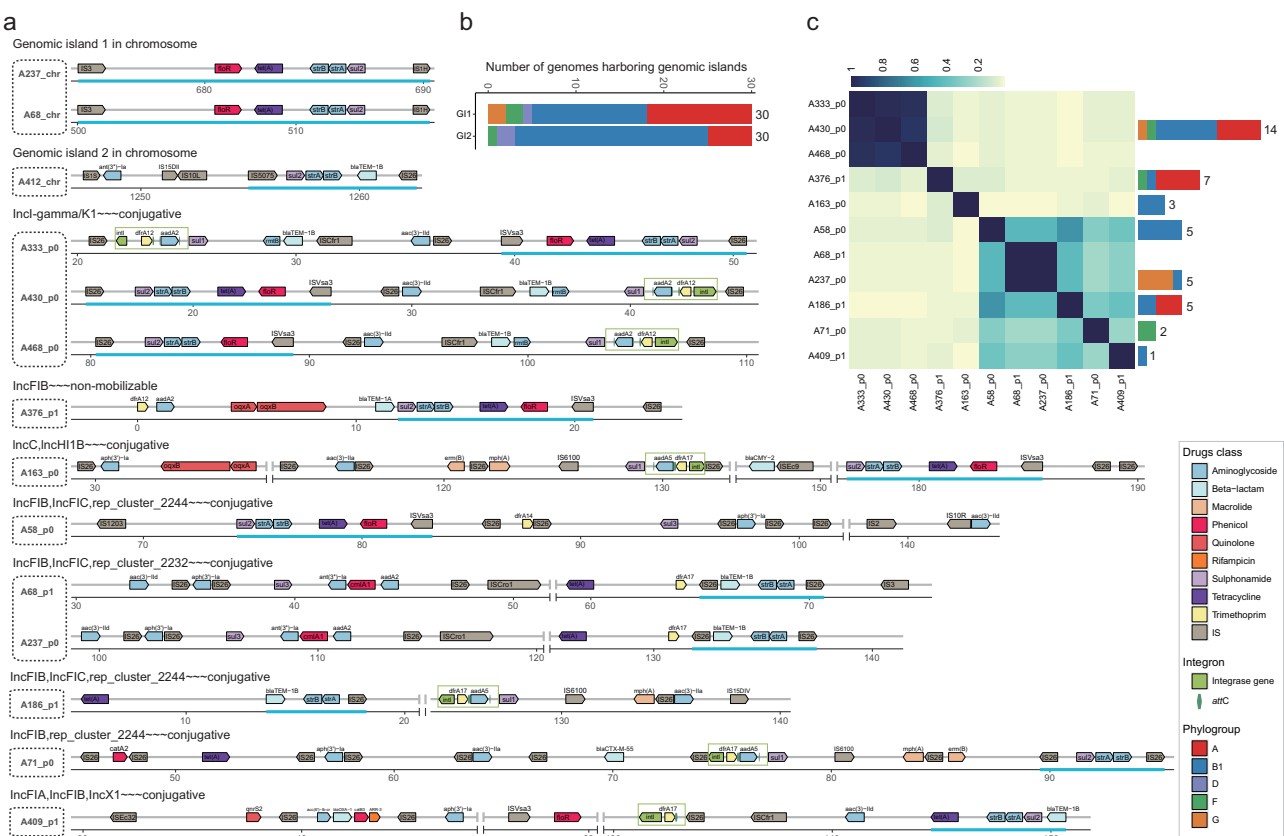

**Fig. 4 | Genetic contexts of vectors harboring ARGs with high co-occurrence frequency in 20 complete genomes. a** Gene arrow maps illustrating ARGs and their neighboring ISs. The direction of the arrows corresponds to the strand where the genes are situated. Right-pointing arrows represent genes on the forward strand, while left-pointing arrows indicate genes on the reverse strand. Different colors of the arrows signify distinct gene types. The start and end positions of the genes within their respective molecules are visually depicted along the x-axis, measured in kilobase pairs (kbp). Highlighted blue lines emphasize potential co-transfer units of specific significance, while green boxes represent predicted integrons. The labels on the left side denote the names of each vector, and information about the plasmid's replicon types and mobility (separated by '~~~') is provided above the dashed box. Gene names and IS elements, which conventionally appear in italics, are displayed in regular font for clarity. A412_chr exhibits only a subset of ARGs and ISs. **b** The stacked histogram illustrating the distribution ExPEC isolates for various phylogroups harboring genomic islands. **c** The prevalence of the plasmids in 499 ExPEC genomes. The heatmap illustrates the Jaccard similarity coefficient among pairs of plasmids. The stacked histogram on the left displays the number of genomes for different phylogroups harboring each plasmid. For clusters of highly similar plasmids (such as A333_p0, A430_p0, and A468_p0), we employ the prevalence of the plasmid with the shortest sequence among them as a representative measure for that type of plasmid. Source data are provided as a Source Data file.

---

*dfrA12*, and *tet(A)* was identified with two copies situated on separate plasmids. Additionally, we also identified integrons within the 20 complete genomes. A total of 16 integrons were detected, all of which were located within the plasmid genomes (Supplementary Data 11). Furthermore, we found that these integrons generally contain two ARG cassettes: *aadA2/aadA5* and *dfrA12/dfrA17*. It is noteworthy that we observed these integrons closely associated with IS*26* (Fig. 4a). This suggests that the accumulation and dissemination of specific ARGs in swine ExPEC were facilitated by the interplay between integrons and transposons.

When the co-location of the ARGs were considered, we found that the five genes *sul2*, *strA*, *strB*, *tet(A)*, and *floR* were clustered together on both chromosome and plasmid (Fig. 4a). The ARG cluster on the chromosome is delimited by IS*3* and IS*1H*, forming Genomic Island 1 (GI1), while on the plasmid, it is framed by IS*Vsa3* and IS*26*, which are associated with IncI-gamma/K1 plasmid. Moreover, we observed that GI1 was detected in 30 isolates (Fig. 4b), while the IncI-gamma/K1 plasmid was present in 14 isolates (Fig. 4c). It's worth noting that these isolates were distributed across phylogroups A, B1, D, F, and G. Another ARG cluster located on the chromosome, is consisting of *sul2*, *strA*, *strB*, and *bla*TME-1B, with IS*5075* and IS*26* serving as flanking elements, collectively forming Genomic Island 2 (GI2). It was identified in 30 isolates, primarily from phylogroup B1. Regarding plasmids, two

similar gene clusters were observed, with both of these gene clusters being associated with IS*26*. One contains *strA*, *strB*, and *bla*TME-1B (A68_p1, A237_p0, A186_p1), while the other harbors *strA*, *strB*, and *sul2* (A71_p0). These findings indicate that the transfer of this ARG cluster occurs in a diverse manner, facilitated by transposable elements (TEs) among chromosomes and also by conjugative mobilization through plasmids. Moreover, in addition to the ARG clusters, these conjugative plasmids carried an array of diverse ARGs, including *aac(3)-IId*, *mph(A)*, *cmlA1*, *sul3*, and *dfrA17*, which confer resistance to antibiotics belonging to aminoglycosides, macrolides, phenicols, sulfonamides, and trimethoprim. These findings highlight the plasticity and heterogeneity of the gene clusters associated with *strA* and *strB*, enabling *E. coli* to combat a broad spectrum of antibiotic agents.

### Plasmids carried by swine ExPEC have the potential to transfer ARGs against important antibiotics used in human treatment

We identified potential horizontal gene transfer facilitated by the plasmid by discerning similarities between the plasmids found in ExPEC carrying ARGs and those identified in bacteria from other hosts, especially human (Fig. 5). Among the 20 isolates selected for sequencing of closed genomes, A376 and A382 showed resistance to carbapenems. Two plasmids belonging to IncX3 (A382_p1) and IncFIA/FIC (A376_p2) harbored *bla*NDM-1 and *bla*NDM-5, respectively (Fig. 5a).

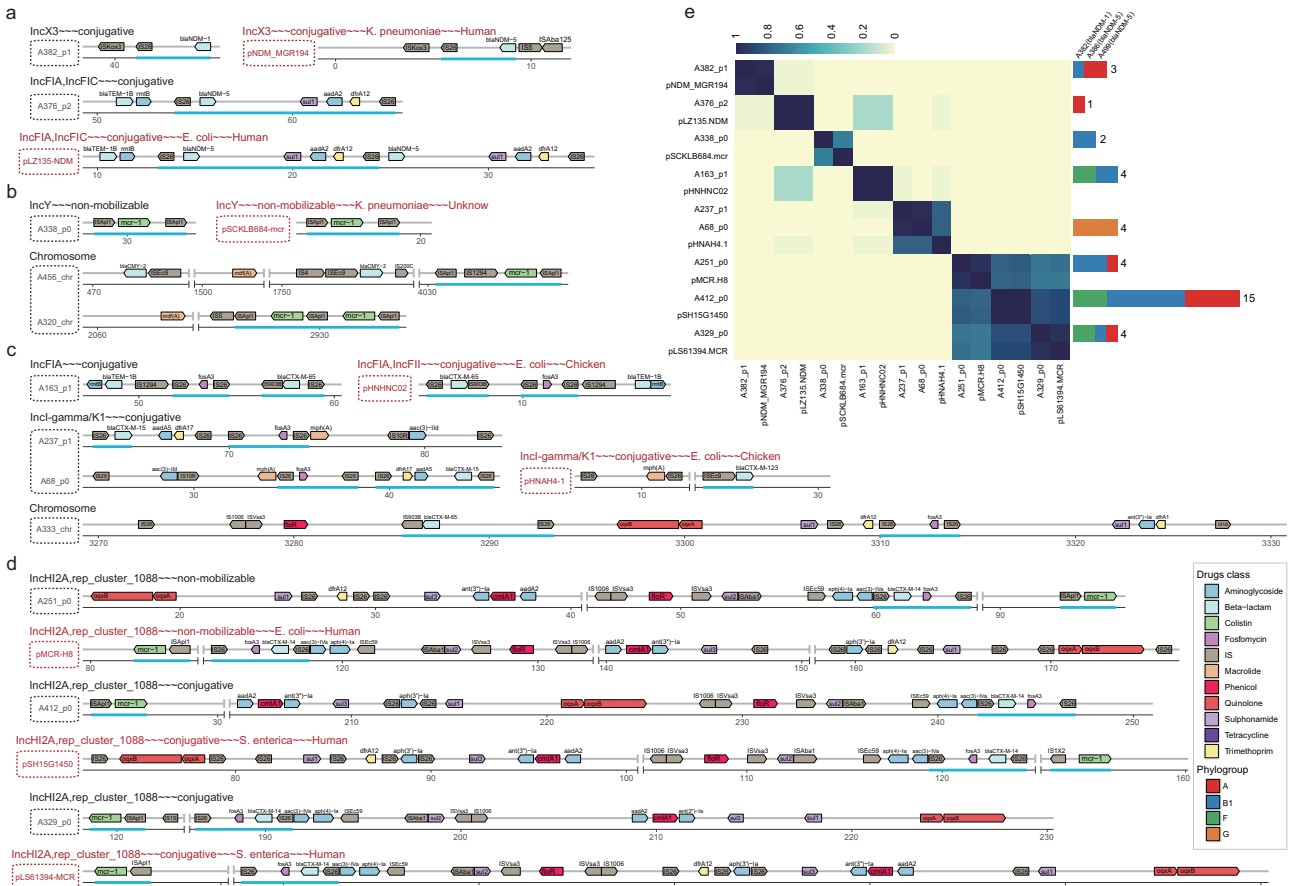

**Fig. 5 | Genetic contexts and comparisons of vectors containing ARGs against crucial antibiotics across various strains from different hosts.** Gene arrow maps illustrating the presence of ARGs: $bla_{NDM}$ (Panel **a**), $mcr-1$ (Panel **b**), co-existence of $fosA3$ and ESBLs (Panel **c**), and co-existence of $mcr-1$, $fosA3$, and ESBLs (Panel **d**). The meaning represented by the depicted elements is similar to that of the elements in Fig. 4a. Black labels for vector names denote plasmids from our study, while each red label corresponds to plasmids from public databases that are most similar to the respective black-labeled plasmid. Information regarding the replicon types, mobility, species name of the strain, and the host of the strain (separated by '---') for public plasmids is provided above the dashed box. A333_chr only shows a partial representation of ARGs and ISs. Panel **e**: The similarity between plasmids from ExPEC and those from bacteria originating from other hosts, as well as the prevalence of plasmids within the ExPEC context. The meaning represented by the depicted elements is similar to that of the elements in Fig. 4c. Due to partial similarities and some distinctions among plasmids A251_p0, A412_p0, and A329_p0, and in accordance with the principle of plasmid incompatibility, it is not possible for a bacterium to simultaneously possess two such plasmids. Consequently, if the sequencing data of a bacterium mapped to the genomes of these three plasmids, each exhibiting a coverage exceeding 90%, we chose the result with the highest coverage. Source data are provided as a Source Data file.

Genomic analysis revealed a high similarity between plasmid A382_p1 and plasmid pNDM_MGR194 found in the human pathogenic bacterium *Klebsiella pneumoniae*[46] (Fig. 5e). The main difference lies in the resistance gene carried by the pNDM_MGR194 and pNDM_MGR194-like plasmids, which is $bla_{NDM-5}$. However, the insertion sequences on one side of these plasmids are the same. Three strains belonging to phylogroups A and B1 were found to possess plasmid A382_p1 in our swine ExPEC collection (Fig. 5e). Among these isolates, A386 and A499 were found to harbor the antibiotic resistance gene $bla_{NDM-5}$, consistent with the observations made in pNDM_MGR194. For the $bla_{NDM-5}$ gene, situated on plasmid A376_p2, it was found to be clustered with other ARGs conferring resistance to beta-lactams, aminoglycosides, sulfonamides, and trimethoprim. These ARGs on the plasmid are flanked by IS*26*, forming a potentially transposable unit. The plasmid A376_p2 exhibits significant similarity to the plasmid pLZ135-NDM in human *E. coli*, which contains two copies of this transposable unit. These indicate the potential transmission of $bla_{NDM}$ genes between swine *E. coli* and human pathogens through plasmids, highlighting the variability of the $bla_{NDM}$ gene on conjugative plasmids in the swine ExPEC.

Seven copies of the colistin resistance gene $mcr-1$ were identified in the genomes of six isolates, with four copies located on four plasmids (A338_p0, A251_p0, A412_p0, A329_p0) and three copies found on two chromosomes (Fig. 5b and d). In all of these cases, the insertion sequence IS*Apl1* was found in close proximity to the $mcr-1$ gene, as reported previously[47,48]. Plasmid A338_p0 exhibits moderate genomic sequence similarity to plasmid pSCKLB684-mcr from *K. pneumoniae*. Plasmid A251_p0, A412_p0, and A329_p0 exhibit a high degree of similarity in terms of gene content; however, the latter two are capable of conjugative transfer, while the former is non-mobilizable. Plasmid A412_p0 and A329_p0 share a high level of genomic sequence identity with plasmids pSH15G1450 and pLS61394-MCR, both originating from *Salmonella enterica* isolated from children with intestinal infections in China[49,50]. And plasmid A251_p0 is homologous to plasmid pMCR-H8 from ESBL-producing *E. coli* H8, isolated from the mink farmer in Shandong Province of China[51]. Importantly, the A251 strain possesses zoonotic potential based on the cgMLST results (Supplementary Data 2). Besides $mcr-1$, these plasmids harbor a diverse array of additional ARGs conferring resistance to crucially important antibiotics for human treatment, including $bla_{CTX-M-14}$ (cephalosporin), $floR$ (florfenicol), $fosA3$ (fosfomycin), and $oqxA$ and $oqxB$ (quinolone). Analysis of plasmid prevalence showed that the conjugative plasmid A412_p0 and A329_p0 were present in 15 and 4 swine ExPEC strains, respectively, whereas the non-mobilizable plasmid A251_p0 could be found in 4 isolates (Fig. 5e).

In addition to the *fosA3* genes that co-locate with the *mcr-1* genes, we identified an additional four *fosA3* genes on the 20 genomes analyzed. Among these, one gene was located on the chromosome, while the remaining three were found on three different plasmids belonging to two distinct families, namely IncFIA and IncI-gamma/K1 (Fig. 5c). Furthermore, we observed that on these plasmids, two ESBL genes ($bla_{CTX-M-15}$ and $bla_{CTX-M-65}$) were found in close proximity to *fosA3*. Additionally, all *fosA3* genes and ESBL genes were associated with IS*26*. Notably, among the three plasmids coexisting with *mcr-1*, the gene $bla_{CTX-M-14}$ and *fosA* constitute a potential co-transfer unit with two IS*26* elements serving as flanking sequences (Fig. 5d). Comparative genomic analysis showed that plasmids A163_p1 was highly similar to plasmids pHNHNC02 from a chicken-derived *E. coli* isolate[52] (Fig. 5e). Significantly, plasmids A237_p1 and A68_p0 have a greater diversity of ARGs compared to their closest counterpart, plasmid pHNAH4-1 from a ceftazidime-resistant *E. coli* strain isolated from chicken feces in China which had no fosfomycin resistance gene[53]. This finding suggests that these two plasmids represent a novel type carrying both *fosA3* and ESBL genes. Taken together, we observed that plasmids carrying the *fosA3* gene have a wide host range, being found in different *Enterobacteriaceae* members of swine, poultry, and human origin.

## Discussion

Our large-scale population genomic analysis of 499 swine ExPEC isolates collected from China revealed that more than 80% belonged to phylogroups B1 and A. In pig farms worldwide, encompassing regions such as Australia and Poland, the population structure of *E. coli* isolated from the feces of healthy pigs was also predominantly characterized by phylogroups B1 and A[54–56]. Moreover, several prevalent STs identified in our dataset coincide with those observed in healthy pig intestinal *E. coli*[57], such as ST10, ST58, ST88, ST101, ST117, and ST744. These findings further support the inference that ExPEC originate from fecal sources and exist asymptomatically within the intestinal tract[10]. Additionally, these two phylotypes have been reported to be associated with the majority of enterotoxigenic *E. coli* (ETEC), and many *E. coli* strains from animals, birds, and the environment have been found to belong to these two groups[58–60]. However, our survey of virulence factors enables us to distinguish these *E. coli* strains from ETEC, as they exhibit a very low frequency of ETEC toxin genes. Although only a small number of ExPEC strains contain toxins typical of enteric *E. coli* pathotypes, caution must be exercised regarding these potential hybrid pathotypic strains. For instance, ETEC strains, mainly including F4+ and F18+ *E. coli*, were often the cause of post-weaning diarrhea (PWD), which incurs significant economic costs for the pig industry[61]. F4-positive ETEC isolates causing PWD commonly belonged to ST100, while F18-positive isolates were assigned to diverse STs, including ST23, ST10, and ST155[62]. In our study, all three F4-positive ExPEC/ETEC strains belonged to ST100, and one F18-positive ExPEC/ETEC strain belonged to ST155. This caution arises from the potential of hybrid pathotypic *E. coli* to cause more severe illnesses and exhibit higher levels of resistance compared to non-hybrid pathotypes[63–65]. Furthermore, our investigation illustrated that swine-derived hybrid pathotypic strains displayed heightened antibiotic resistance.

On the other hand, compared to human-derived ExPEC, swine-derived ExPEC displayed a distinct population structure, with human strains often belonging to phylogroups B2 and D[7,8,66]. The most prevalent STs of human ExPEC, such as ST131, ST69, ST95, ST648, ST73, ST393, and ST12[1,60], were not commonly found in our dataset. However, a small proportion of swine ExPEC belonged to these ST types, including ST131, ST69, ST95, and ST648. These strains carried ESBL genes, and the majority of them also harbored genetic determinants of fluoroquinolone resistance. A study has indicated that swine *E. coli* belonging to ST131 posed a potential zoonotic risk[67]. In our

investigation, swine ExPEC belonging to ST131 and ST648 were also found to have zoonotic potential. Additionally, there were several prevalent STs shared by both human and swine ExPEC. The prevalent STs in our collection, including ST410, ST354, ST88, and ST23, were also reported to be among the top 20 human ExPEC STs[1]. ExPEC strains of ST410, which are resistant to third-generation cephalosporins and carbapenems, have raised significant concerns recently[68,69]. ST354 represented predominant ST of fluoroquinolone-resistant *E. coli* originating from canine fecal and extraintestinal infections, with certain strains within this ST type carrying ESBL genes[70]. In our study, swine ExPEC of ST354 also exhibited similar characteristics: the hybrid pathotype, resistance to fluoroquinolones, and the presence of ESBL genes. Swine ExPEC of the ST10 complex within phylogroup A exhibited a high degree of diversity, including ST774, ST167, ST617, ST10, and several other sequence types (STs) with fewer than 5 members. Among them, ST10, ST167, and ST617 have been reported as major STs associated with human ExPEC infections[1]. Similarly, our study's cgMLST analysis revealed that some ExPEC strains in ST167, ST617, or ST744 were sourced from humans. Other STs were once reported to be associated with human ExPEC, including ST117 and ST58. ST117 from phylogroup G has been proposed as an emerging multidrug-resistant foodborne ExPEC[71]. These findings suggest that, despite differential transmission dynamics of ExPEC between swine and humans, the potential transmission of ExPEC between humans and swine may serve as a significant source of human ExPEC infections. This highlights the importance of understanding and monitoring the dynamics of ExPEC populations in both animal and human settings.

Antibiotic resistance has emerged as a pressing concern in bacteria from animals and their respective environments[72–74]. Our collection of *E. coli* strains exhibited extensive resistance to various classes of drugs, surpassing the severity of resistance levels reported previously[43,44,74]. Particularly noteworthy was the heightened prevalence of MDR within swine ExPEC. This phenomenon may be attributed to the enduring use of antibiotics in China's livestock industry over the preceding decades[42]. The antibiotic resistance profiles suggested these ExPEC isolates not only have obtained strong resistant abilities to the frequently used veterinary antibiotics, but also have developed resistance to some important clinical antibiotics. Moreover, it is noteworthy that over 50% and 40% of the tested *E. coli* strains respectively exhibited resistance to third- and fourth-generation cephalosporins, which have been utilized in the treatment of a broad range of severe infectious diseases in humans[75,76]. In 2020, China implemented a ban on the use of antibiotics in feed additives[77], and strict monitoring of the types and quantities of antibiotics used to treat animal diseases is now in place.

The prevalence of unique combinations and the co-transfer of multiple resistance determinants underscores the potential for the emergence and spread of MDR strains. The co-selection of multiple ARGs complicates the transmission and accumulation of these genes, as the presence of one antibiotic is sufficient to sustain (select for) all antibiotic resistance genetic determinants due to their co-location on the plasmid[78,79]. In our study, the co-occurrence network of ARGs and the genetic contexts of complete plasmids suggested the presence of diverse patterns in the coexistence of ARGs. This information is valuable for preventing the spread of additional ARGs that coexist with genes selected by widely used antibiotics in livestock. Furthermore, we successfully resolved the co-transfer of multiple ARGs with a high co-occurrence frequency and elucidated how these genes achieve widespread presence in this population through the utilization of insertion sequences and conjugative plasmids. Our findings shed light on the mechanisms driving the widespread presence of ARGs in this population, providing a basis for implementing targeted interventions and control measures to uphold antibiotic efficacy and safeguard livestock health.

*E. coli* has the ability to acquire resistance genes from other bacteria, while also being capable of transferring its own resistance genes to other bacterial strains[80]. The dynamic exchange of resistance genes underscores the potential for *E. coli* to play a pivotal role in the dissemination of antibiotic resistance among bacterial populations in both animal and human hosts. Plasmids have been identified as the most crucial vectors of ARGs for the dissemination events[28,81,82]. Our findings showed the effective transmission of the carbapenemase-encoding gene $bla_{NDM}$ between swine ExPEC, and human pathogens like *K. pneumoniae* and *E. coli* through specific plasmid types. Carbapenem-resistant *Enterobacterales* (CRE) infections have become a major concern in the global effort to combat infections[83]. In recent years, there has been an increasing global transmission of CRE, and this situation has been further influenced to some extent by the spread of COVID-19[84]. Our study provided valuable insights that could contribute significantly to CRE prevention and control efforts. Additionally, colistin resistance is a pressing issue globally, primarily because colistin is considered as a last-resort medication for tackling drug-resistant bacterial infections[85]. Multiple studies have provided compelling evidence of the extensive dissemination of the colistin resistance gene *mcr-1* through conjugative plasmids in various settings, including the livestock farms, human populations and natural environment[86–88]. Our study unveiled the combined action of conjugative and non-mobile plasmids in perpetuating *mcr-1* prevalence, highlighting the persistent challenge posed by colistin resistance despite regulatory interventions. Colistin resistance genes encompass various types, with *mcr-1* and *mcr-3* having been disseminated worldwide[48,89]. It is notable that strains simultaneously carrying both *mcr-1* and *mcr-3* have been isolated from various sample types, encompassing cattle in Spain and Italy[90,91], as well as pigs, poultry, and clinical samples in China[92–94]. Our study describes two swine ExPEC strains concurrently harboring *mcr-1* and *mcr-3*. While this occurrence is not prevalent, vigilance regarding the risk of its accumulation is warranted. The prevalence of extended-spectrum β-lactamases (ESBLs) poses a significant challenge to the utilization of β-lactam antibiotics in human medicine, with CTX-Ms being the predominant ESBL family worldwide[95]. The limited availability of new antibiotic agents has prompted a re-evaluation of previously used antibiotics, such as fosfomycin, as potential treatment options for multidrug-resistant bacteria, particularly those producing extended-spectrum β-lactamases (ESBLs) and carbapenemases within the *Enterobacteriaceae* family[96]. However, several studies have revealed a concerning trend of co-localization of ESBL genes and *fosA* genes on plasmids, presenting an additional challenge in the treatment of *Enterobacteriaceae* infections[97–99]. Our investigation revealed the co-localization of *fosA3* with multiple ESBL genes on diverse plasmids, aiding in the tracing of the dissemination of these ARGs. In the context of One Health, these findings highlight the need for continued vigilance and comprehensive strategies to monitor and address the potential reservoirs and transmission routes of ARGs, ensuring effective measures are in place to mitigate the risk of their persistence and spread in both animal and human populations.

## Methods

### ExPEC isolates collection and DNA extraction

The ExPEC isolates were collected from diseased tissue samples of swines exhibiting various extraintestinal infection clinical symptoms by the Animal Disease Diagnostic Laboratory at Huazhong Agricultural University during 2011–2017. Initially, swines suspected of having extraintestinal infections were identified through clinical symptoms such as lying on the side, abdominal breathing, shaking, and lameness. These assessments were conducted by trained veterinarians and animal health professionals. Tissues showing pathological changes from swines that died with these symptoms were collected for further analysis, including lung, liver, brain, and heart. The tissue sampling was

conducted using aseptic instruments, with operators wearing personal protective equipment. Samples were collected swiftly, sealed in sterile containers, and refrigerated to preserve their integrity. Detailed records of the diseased tissues, sample collection times, and the geographical locations of the pig farms were documented. The samples, along with the documented records, were transported to the Animal Disease Diagnostic Laboratory at Huazhong Agricultural University. The laboratory was notified in advance of the incoming samples, and a communication channel was established between the field veterinarians and the laboratory personnel to ensure seamless transfer and receipt of the samples. Upon receiving the samples, the laboratory conducted further diagnostic tests to confirm the presence of ExPEC. The results were then reported back to the originating veterinarians and farms for further action.

Given that our study involved isolating *E. coli* from the tissues of deceased animals, the experiment received ethical approval from the Animal Management and Ethics Committee of Huazhong Agricultural University (No. HZAUSW-2017-006). Moreover, in handling tissues from deceased animals, we strictly adhere to the "Technical Specification for the Harmless Treatment of Dead Animals" issued by the Ministry of Agriculture of the People's Republic of China (https://www.gov.cn/gongbao/content/2013/content_2547154.htm).

These samples were obtained from pig farms in various regions of China, including the provinces of Hubei, Hunan, Henan, Jiangxi, Anhui, Zhejiang, Guangdong, and Jiangsu. The collected materials were inoculated onto MacConkey agar plates and incubated at 37 °C for 12 hours. Colonies with characteristics indicative of *E. coli* were selected and subcultured onto a general-purpose agar medium, followed by further incubation at 37 °C for 12 hours to achieve purification. Each *E. coli* isolate was confirmed using 16 S rRNA gene PCR. Further identification of ExPEC involved PCR utilizing specific oligonucleotide primers to amplify five virulence genes: *papA* and/or *papC* (P fimbriae), *sfa* or *foc* (S or F1C fimbriae), *afa* or *dra* (Dr-antigen-binding adhesins), *iutA* (aerobactin receptor), and *kpsMT II* (group 2 capsular polysaccharide units). ExPEC strains were characterized by the presence of at least two of the aforementioned five markers. The primer sequences are provided in Supplementary Data 12.

The ExPEC strains were isolated from diseased pigs sent to our laboratory from various regions across China, forming a comprehensive sample pool. This strain pool served as our sampling frame, which included all ExPEC isolates received from 2011 to 2017. Each year, we selected 50-90 strains for genome sequencing based on practical considerations, including the total number of isolates received annually and the costs associated with sequencing and analysis, resulting in a total collection of 499 strains. A random number generator was used to select isolates from the pool each year, ensuring that each isolate had an equal chance of being included in the study. The genomic DNA was extracted from all ExPEC isolates using a commercial DNA Kit (TIANGEN, Beijing, China). The quality and concentration of the bacterial genomic DNA were assessed through electrophoresis on a 1% agarose gel, and analysis was conducted using a NanoDrop2000 system (Thermo Scientific, Waltham, MA, USA) and a Qubit 4 Fluorometer (Thermo Scientific, Waltham, USA).

### Antibiotic susceptibility testing

The MIC values of various antibiotics (Supplementary Data 13) were employed to evaluate the antibiotic susceptibility of ExPEC isolates, and its testing methods and breaking points followed the recommended microbroth dilution protocol (CLSI M100, 28th Edition) of the Clinical & Laboratory Standards Institute (CLSI, United States). The collection of MIC values mainly relied on the BD Phoenix™ 100 automatic microbial identification and susceptibility analysis system. A total of 20 tested antibiotics belonged to six broad categories, including aminoglycoside (amikacin and gentamicin), beta-lactam (penicillin: amoxicillin clavulanic acid, ampicillin, aztreonam,

piperacillin, sulbactam ampicillin, and tazobactam piperacillin; first-generation cephalosporin: cefazolin; third- or fourth-generation cephalosporin: cefepime, cefotaxime, and ceftazidime; carbapenem: imipenem and meropenem), chloramphenicol (chloramphenicol), quinolone (ciprofloxacin, levofloxacin, and moxifloxacin), sulfonamide (sulfamethoxazole trimethoprim), tetracycline (tetracycline). Three replicates were set for each antibiotic test. *E. coli* ATCC 25922 and ATCC 35218 strains were used for quality control.

### Short-read assembly and genome annotation
All genomes of the studied *E. coli* isolates were sequenced through the Illumina HiSeq 2500 platform. Genome assembly and annotation were performed using modules from the comparative genomic analysis platform (PGCGAP)[100] developed by us previously. The raw short-reads generated by Illumina platform were initially evaluated using FastQC v0.11.9 (https://www.bioinformatics.babraham.ac.uk/projects/fastqc/) and then processed with Trimmomatic v0.38[101] to remove adapters and low-quality reads. The resulting high-quality reads for each strain were assembled using SPAdes v3.14.1[102] (parameters: -k 21,33,55,77 --careful). The draft genomes were annotated by Prokka v1.14.6[103] with the following parameters: --genus Escherichia --species coli --gram neg --usegenus.

### MLST analysis, serotype prediction, VFs and ARGs identification
We employed SRST2 v0.2.0[104] to predict sequence type of each strain based on short-read sequences. ExPEC serotype prediction from assembled genome data was performed using ECTyper v1.0.0[105]. VFs and ARGs were identified based on genome sequences through ABRicate v1.0.1 (https://github.com/tseemann/abricate) using Ecoli_VF database (https://github.com/phac-nml/ecoli_vf) and ResFinder database[106], respectively (minimum identity 90%, minimum coverage 80%). Chromosomal point mutations contributing to antibiotic resistance were detected using ResFinder v4.1.5[107] with the "--point" parameter, based on the PointFinder database[108].

### Pangenome analysis and phylogenetic analysis
Pangenome analysis was conducted using Roary v3.13.0[109] with the following parameters: -i 85 -e --mafft. This resulted in the creation of a multi-FASTA alignment of 2,241 core genes. Then we used SNP-sites v2.5.1[110] to extract the single nucleotide polymorphisms (SNPs) from the core genome alignment, resulting in a total of 208,012 SNPs being identified. The Maximum Likelihood (ML) phylogenetic tree was constructed using RAxML v8.2.12[111] with the GTR + GAMMA model and 1000 bootstrap replicates.

### cgMLST analysis
We performed cgMLST analysis for each strain using cgMLSTFinder v.1.2.0 (https://bitbucket.org/genomicepidemiology/cgmlstfinder) with the Enterobase Escherichia/Shigella cgMLST v1 scheme. The results provided the closest cgST type along with the number of matching alleles for each strain. Based on this, we further filtered the results with a threshold of ≤10 allele differences compared to the closest cgST type. Consequently, in the filtered results, the ExPEC strains assigned with cgST types shared a common clonal origin with strains in the Enterobase database belonging to the same cgST types. Information regarding the source of public isolates (e.g., bovine, human) was extracted from the "Source Type" and "Source Details" columns in EnteroBase.

### Long-read assembly
Twenty representative isolates from different phylogroups and STs were selected to obtain complete genomes. These genomes were sequenced through the Oxford Nanopore Technologies (ONT) MinION sequencer. Guppy v5.0.16 was employ for ONT basecalling, and only reads that passed Guppy's quality filters were retained (mean_qscore_template ≥7). Subsequently, we utilized Filtlong v0.2.0 (https://github.com/rrwick/Filtlong) to remove the worst 5% of long-reads. Finally, summary statistics of filtered long-reads were evaluated using NanoPlot v1.29.1[112]. The results indicated that the mean read quality of these samples was around 9, and the mean read length was around 20 kbp (Supplementary Data 14). Additionally, the mean size of each sample yielding filtered data reached 1168 Mbp, with a minimum of 669 Mbp, reflecting a sequencing depth of at least 100x. De novo assembly of filtered long-reads was carried out using Flye v2.8.3-b1695[113]. The resulting genome sequences underwent two rounds of polishing with Pilon v1.23[114], using Illumina reads.

### Annotation of complete genomes and comparative analysis of plasmids
The replicon family and transferability of 43 plasmids were predicted using MOB-typer module in the MOB-suite v3.0.3[115]. Additionally, the 'mash_nearest_neighbor' entry in the tool's results as the basis for the most similar matches to plasmids in public databases, except for plasmid A381_p1. In the case of A381_p1, its closest matches were determined from the literature[46], considering gene content and replicons. The annotation of ARGs in the chromosome and plasmid genomes was carried out as previously described. Insertion sequences (ISs) were identified using ABRicate v1.0.1 (https://github.com/tseemann/abricate) with ISfinder database[116] (minimum identity 90%, minimum coverage 80%). Integrons were detected using IntegronFinder v2.0.3[117] with the following parameters: --local-max --funcannot. The sourmash v3.3.0 program[118] was employed, utilizing a k-mer size of 31, to calculate Jaccard indices between pairs of plasmids.

### Detection of Genomic Islands on the chromosomes
Detecting complete genomic islands in genomes assembled from short-read data is challenging, as draft genomes exhibit significant fragmentation. Hence, we applied the following criteria to ascertain the existence of a genomic island in a draft genome. First, the presence of all ARGs associated with that genomic island must be confirmed in the genome. Second, for the identification of a partial genomic island—such as for the detection of Genomic Island 1—*strB*, *strA*, *sul2*, IS1H in the draft genome should be located on the same contig. Similarly, for the detection of Genomic Island 2—*strB*, *strA*, *sul2*, IS5075 in the draft genome must be present on the same contig. The above-mentioned process was accomplished using a custom Python script (available at https://github.com/xdli009/Swine_ExPEC_genomes_analysis/tree/main/Detection_Genomic_Islands).

### Mapping short reads to plasmid sequences
Mapping short sequence reads from strains that were not subjected to long-read sequencing to complete plasmid sequences was employed to determine the distribution of these plasmids across all ExPECs. We utilized BWA v0.7.17-r1188[119] to align short reads from each strain to the plasmid genomes (minimum mapping quality 30) and computed coverage for sequence alignments across each plasmid using bedtools v2.30.0[120]. To confirm the presence of a specific plasmid within an ExPEC strain, two criteria must be met: firstly, the alignment coverage should be at least 90%, and secondly, the assembled genome's predicted ARGs must encompass a minimum of 80% of the ARGs present in that plasmid.

### Visualization
We employed ggplot2 v3.3.6 (https://ggplot2.tidyverse.org.) and UpSetR v1.4.0[121] packages in R (available at https://github.com/xdli009/Swine_ExPEC_genomes_analysis/tree/main/Visualization) to visualize phenotype and genotype data. Phylogenetic trees were visualized by iTOL v5[122]. The co-occurrence network of ARGs was generated using Cytoscape v3.8.0[123]. The ARGs and ISs arrow maps for the complete genomes were created using gggenes v0.4.1 (https://github.com/wilkox/gggenes). Heatmaps to visualize Jaccard similarity matrix of plasmid genomes were generated using pheatmap v1.0.12

(https://CRAN.R-project.org/package=pheatmap). We used Adobe Illustrator (http://www.adobe.com/au/products/illustrator.html) for figures retouching.

## Statistics & reproducibility

In this study, no statistical analyzes were conducted. Our approach was based on direct observation, descriptive analysis, or other non-statistical methods suitable for the objectives of our research. Biological samples were collected randomly without any prior information about their antibiotic resistance phenotypes. To ensure the reproducibility, each antibiotic susceptibility test was performed in triplicate. Only results that were consistent across all three replicates were considered valid and included in our analysis.

## Reporting summary

Further information on research design is available in the Nature Portfolio Reporting Summary linked to this article.

## Data availability

All clean short-read and long-read sequence data, draft genomes, and complete genomes generated in this study have been deposited in the NCBI under BioProject ID PRJNA1044843. BioSample accession number for each strain are provided in Supplementary Data 1. Source data are provided with this paper.

## Code availability

All custom Python and R codes have been uploaded to GitHub: https://github.com/xdli009/Swine_ExPEC_genomes_analysis.git.

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

## Acknowledgements

This work was funded by the National Key Research and Development Program (2022YFD1800901, 2021YFD1800402), the China Agriculture Research System of MOF and MARA (CARS-35), the Walmart Food Safety Collaboration Center of Walmart Foundation (Project # 61626817), the National Natural Science Foundation of China (31970003). Dr. Michael Gänzle are acknowledged for critical feedback on the manuscript.

## Author contributions

J.Z. and C.T. designed the study; C.T., Y.Z., Y.L., X.W., and Z.P. conducted sample collection and laboratory experiments; X.L., H.H., and T.W. conducted bioinformatic analyzes; X.L. and J.Z. drafted the manuscript; C.T., M.S., and H.C. revised the manuscript. All authors reviewed and approved the final version of the manuscript.

## Competing interests

The authors declare no competing interests.
