## [Peer Review File · Nature Communications]

Population structure and antibiotic resistance of swine extraintestinal pathogenic *Escherichia coli* from ChinaREVIEWER COMMENTS

Reviewer #1 (Remarks to the Author):

To Whom It May Concern:

While the role of poultry in the occurrence of human extraintestinal pathogenic *E. coli* infections has been explored in many studies, few have examined the link between those from swine. This research by Li et al is well executed, well communicated and is a considerable contribution to the field.

There is a great need to identify emerging pathogens which may threaten human and animal health as well as those which pose risks to industry and the environment. I congratulate the authors on their efforts, the quality of their work and their contribution to the literature. The strains described in this article are concerning, particularly given their known role in human infections (at the ST level) and their alarming resistance profiles.

Generally speaking, the methodology is sound, though future studies carried out by the authors would benefit from refreshed methodology (described later in more detail), and deposition of code in a publicly available repository (i.e. on github) would improve reproducibility. I appreciate the breadth and quality of the data visualisation. While the authors have clearly reviewed a great deal of literature, some key articles detailing established links between swine and human sourced *E. coli*, and other relevant findings, were not referenced by the authors. In general, conclusions drawn by the author are well substantiated, however I have included below some relevant commentary and specific papers which the authors seem to have missed during their research which I think can improve their article.

This manuscript provides compelling evidence that the relentless use of first-generation antibiotics has led to the evolution of genetic platforms that remain at the core of complex resistance regions that are found in these invasive ExPEC populations with genes encoding resistance to clinically important antibiotics being added (but not replacing) to these platforms. This is a worrying trend and a true One Health threat.

I will include more commentary below.

Kind regards,

Dr Max Cummins

More thoughts:

- Title: perhaps Diversity "and" population structure is more precise
- Aim on line 56 is very broad and beyond the scope of the current manuscript in its current form.
- AbuOun et al, 2020 (<https://doi.org/10.3389/fmicb.2020.00861>) details (in Table 7) that many of the same STs identified in the present article are also observed in the gut of healthy pigs. This might indicate that the ExPEC in these swine have a gastrointestinal origin.
- High carriage rates of B1 and A phylogroup *E. coli* in the swine gut has been reported, which is unsurprising given commercial pig breeds are globally distributed and diets are largely consistent across major swine producing countries (although AMR stewardship practices vary). The discussion would benefit from more detail in this context. The following papers detail this:
 - o Bonvegna et al, 2022; <https://doi.org/10.3390/antibiotics11121774>
 - o Zingali et al, 2020; <https://doi.org/10.3390/microorganisms8060843>
 - o Reid et al, 2017; <https://doi.org/10.1099/mgen.0.000143>
- Authors do not appear to comment on the pathogenesis of these lung infections. Are pigs inhaling faecal dust (as is common in poultry operations)? This is important context for assessing the risk posed by these pathogens to human and animal health
- I encourage the authors to provide data on the carriage of class 1 integrase gene (*intI1*), as these loci play critical roles in promoting multidrug resistance and in tandem with IS26 elements. These *intI1* genes are often truncated by IS26; their detection will probably require flexible coverage cut-offs. If the role of class 1 integrons in the evolution and dissemination of multidrug

resistance is not clear to the authors, please refer to Gillings et al (<https://doi.org/10.1128/JB.00152-08>).

- While this paper is already bursting at the seams with quality data, large scale SNP analyses would be useful to determine whether isolates closely related to those under study exist in swine elsewhere, or are associated with other environments and hosts. cgMLST-based approaches (<https://github.com/MDU-PHL/Coreugate>), or highly scalable SNP-analyses like SKA (<https://github.com/simonrharris/SKA>) or SKA2 (<https://github.com/bacpop/ska.rust>) would allow the authors to potentially find evidence of clonally related isolates in public databases (e.g. <https://enterobase.warwick.ac.uk/>), from which members of the primary STs identified could be downloaded and compared with those from the present study.

Formatting/typographical/technical/linguistic errors:

- “Diverse population structure among swine ExPEC” seems imprecise/incorrect. Perhaps something like “Swine ExPEC under analysis were phylogenetically diverse” or “exhibited a complex phylogenetic structure”
- IS elements should be partially italicised (e.g. IS26), not fully italicised (e.g. IS26 [incorrect]), nor, as the authors have written, in plain format (IS26 [also incorrect])
- Line 131 – Was this statistically validated? If not, please choose different language

Methodological overview:

Generally speaking, the choice of methodology seems appropriate and well considered, however some of the tools utilised are quite dated and some details are missing relating to parameters utilised for each tool.

Perhaps the most consequential methodological choices are those relating to long-read assembly. The methods chosen in this regard are relatively standardised, however this does not mean that they are necessarily appropriate. The authors may also consider stating why hybrid assembly approaches were not taken given that both short-read and long-read sequence data was available.

It is also not clear what quality control procedures were utilised and whether or not there were a sufficient number of reads to perform long-read only assemblies; “Their length and quality were assessed” (Line 589) – to what degree?

Other general methodological comments

- More information needed on the sections starting line 606: “First, 610 the presence of all ARGs associated with that genomic island must be confirmed in the 611 genome”- Confirmed using which tools?

- Versions of R packages utilised should be included

- The code utilised for data visualisation (i.e. in R) should be provided on github. This is a critical part of maintaining reproducibility and promoting transparency in bioinformatic analyses

- Line 631 – Based on “jaccard indices” – is this to say that jaccard distance based clustering approaches were utilised? Please clarify language here

- I would recommend Bakta (<https://github.com/oschwengers/bakta>) be utilised for future investigations instead. Prokka was released 10 years ago has not been updated for five years (<https://github.com/tseemann/prokka/releases>).

A comparison of Prokka, Bakta and other tools can be found here:

<https://www.microbiologyresearch.org/content/journal/mgen/10.1099/mgen.0.000685>.

- Similarly, Panaroo (<https://github.com/gtonkinhill/panaroo>) (Tonkin-Hill et al, 2020), rather than Roary, might be a wise choice for future analyses (the former released in the last few years and the latter also released around a decade ago with no updates released since 2019)

A comparison of Panaroo, Roary and other tools can be found here:

<https://genomebiology.biomedcentral.com/articles/10.1186/s13059-020-02090-4>

- The authors may also be interested in abritAMR, an ISO-accredited resistance detection tool (<https://github.com/MDU-PHL/abritamr>) (Sherry et al, 2023), which would provide more robust resistance data than that provided by abricate.

- Pilon has been shown to introduce deletion errors by Wick and Holt and I therefore feel its use

should be cautioned: (<https://www.biorxiv.org/content/10.1101/2021.10.14.464465v1>). Wick is well regarded for his expertise in long read assembly – if the authors are not familiar with these pages I encourage them to review them and integrate these learnings into their future long-read assembly endeavours. (<https://github.com/rrwick/Tricycler/wiki>; <https://github.com/rrwick/Tricycler/wiki/Guide-to-bacterial-genome-assembly>).

Reviewer #2 (Remarks to the Author):

The paper reports on a comprehensive investigation of 499 *E. coli* isolates from clinical lesions of pigs from different locations in China. Unfortunately there is no information about how *E. coli* could be linked to the observed lesions including presence of other microbiological agents and it is difficult to designate the isolates as ExPEC on this background. Antimicrobial susceptibility was analysed. The genomes of the isolates were generated and analysed according to best standards including predictions of phylogroup, MLST, virulence genes and antimicrobial resistance genes. A high frequency of multi resistant isolates were predicted with a link to ST410. Based on closed genomes the genetic background for a selection of antibiotic resistance - and potential virulence genes was analysed. Results are discussed in relation to similar *E. coli* populations obtained from humans and some overlaps were found with respect to sequence types and virulence profiles. Unfortunately there is no comparison to *E. coli* isolated from diarrhea of weaned pigs which is the reason for most use of antibiotics. It would have been interesting to know the proportion of ExPEC types isolated from diarrhea and relate the problem to *E. coli* isolated from other lesion types. It would also have been interesting to compare the results to *E. coli* isolated from diarrhea of pigs.

Reviewer #3 (Remarks to the Author):

1. What are the noteworthy results?

The present study shows a comprehensive picture of the putative contribution of swine derived ExPEC to human infections, based on large-scale genome sequencing and antibiotic resistance tests of 499 isolates (central China, 2011-2017).

It is noteworthy the huge genomic diversity found within this swine ExPEC collection in terms of phylogeny, serotypes (200 O:H combinations), genetic virulence or AMR. Important high-risk clones were determined, such as ST23, ST88, ST117, ST131, ST648. It is also of note that >50% of the collection were ESBL producers and >70% showed FQR. To highlight the 406 combinations of different ARGs found among the 492 genomes containing at least 2 ARGs, with an average of 13 ARGs. The top prevalent ARGs were all frequently found in both chromosome and plasmid genomes, such as *sul1*, *sul2*, *floR*, *tet(A)*, *blaTEM-1*, *strA*, *strB*. The authors found some alarming cooccurrences targeting human and veterinary drugs, such as *blaCTX-M-55*, *blaCTX-M-14*, *fosA3*, and *mcr-1*. All *fosA3* genes and ESBL genes were associated with IS26. Notably, among the three plasmids coexisting with *mcr-1*, the gene *blaCTX-M-14*, and *fosA* constitute a potential co-transfer unit with two IS26 elements serving as flanking sequences. Moreover, two of the four swine isolates that showed resistance to carbapenems were selected for sequencing of closed genomes; as a result, two plasmids belonging to IncX3 (A382_p1) and IncFIA/FIC (A376_p2) harbored 362 *blaNDM-1* and *blaNDM-5*, respectively.

2. Will the work be of significance to the field and related fields? How does it compare to the established literature? If the work is not original, please provide relevant references.

The present work is of significance to the field exploring the connection of animal ExPEC to human infections. It is a comprehensive analysis of multidrug resistant mechanisms that reveals potential transmission pathways of ARGs within swine ExPEC and to human pathogens. It thus provides information on a topic that has been little explored to date, and where the control of ExPEC infections is facing increasing complication by the emergence of ARGs.

3. Does the work support the conclusions and claims, or is additional evidence needed?

The conclusions are consistent with the results obtained here. There is no need for further evidence. It is suggested, however, to reinforce some aspects by making some missing information

relevant (see explanation below).

4. Are there any flaws in the data analysis, interpretation and conclusions? Do these prohibit publication or require revision?

There are no flaws in the data analysis, interpretation, and conclusions. However, some additional information and discussion would enrich the results presented here; specifically on the points listed below. So, my suggestion is moderate revision.

-Abstract. Some general phrases could be removed or simplified (e.g. the two first could be "The diversity and antibiotic resistance of animal extraintestinal pathogenic *Escherichia coli* (ExPEC) and its connection to human infections, remain largely unexplored.") in favor of more specific information on ARG co-occurrence or the suggested transmission pathways based on the plasmid analysis. Also some information on the VF should be mentioned.

-Introduction. Line 75-76. It would be interesting to add here information on the role of food-producing animals as a potential source of human ExPEC (including UPEC), as it has been recently proved by different authors including high-risk clones such as ST69, ST95, ST117, ST648 or ST131 (Riley, 2020 [10.1146/annurev-food-032519-051618](https://doi.org/10.1146/annurev-food-032519-051618); Díaz-Jiménez et al., 2021

<https://doi.org/10.1016/j.foodcont.2020.107713>).

Lines 83-85. While this statement is true regarding the in-depth analysis of ARGs, there are some distinct approaches comparing animal ExPEC and those of human infections using different in vitro and in vivo assays that might complete this introduction and/or the discussion (e.g. Zhuge et al. 2020 <https://doi.org/10.1111/tbed.13755>; García et al. 2023

<https://doi.org/10.1016/j.onehlt.2023.100558>).

-Results. Lines 100-104. Here or in Methods: Even if some of this information is partially shown in the Supplementary Data 1, it would be interesting to include data on how many farms, how many geographic regions, and whether isolates are unique or duplicates per animal (outbreaks? isolated cases? Age of the animals/production stage?) to reinforce the idea of the representativeness of the ExPEC collection studied here.

Line 114. Rewrite ". 490 of 499 *E. coli* could be assigned to 75 STs..."

Lines 121-123. This phrase would belong to the discussion section in any case; however, I do not think it is an accurate affirmation without more data because it depends on the geographical area, production status..

Line 134. I suggest removing this line "we also found no certain serotype associated with special isolated time or organ."

80% of the *E. coli* were isolated from lung (I am curious about this: was *E. coli* as the only pathogen present? Was it found in other organs of the animal? Was *E. coli* linked to a respiratory pathology?). "diseased tissues" is the definition of the origin of the isolates (line 100) but there is no mention of the pathologies.

Lines 141-167. I wonder if the authors searched fimbriae associated with ETEC and or STEC pathotypes such as F4, F5, F6, F18, and F41 in the studied collection. If so, please add. According to Supplementary Figure 3, some isolates exhibit hybrid pathotypes (ETEC/STEC/ExPEC, EPEC/ExPEC). This finding is important, even if they are only a few (line 149): to which STs do they belong? Are they ESBL/MDR?

Line 169 "In the past decades, antibiotics were heavily used in poultry and livestock industries" where? in China?. Please, complete.

Line 171. Systematically?

Line 175. What do the authors mean with "unambiguous results". Probably the meaning is: MDR isolates were defined following the Magiorakos et al. (2012) criteria, as those showing acquired non-susceptibility to at least one agent in three or more antimicrobial categories. Please, include the following reference for the definition of MDR: Magiorakos et al. *Clin. Microbiol. Infect.*, 18 (2012), pp. 268-281, [10.1111/j.1469-0691.2011.03570.x](https://doi.org/10.1111/j.1469-0691.2011.03570.x)

Lines 178-181. "strains A376 and A430 exhibiting resistance to all 18 antibiotics tested": 18 out of the 20 antibiotics tested?. The last part of this paragraph should be removed or at least discussed in detail in the following section.

Lines 201-202. The same for the following phrase: "These findings provide a plausible explanation for the popularity of ST410 as the predominant sequence type". There is no consistency in the argument to confirm this. So, I suggest that this should be removed or discuss in detail in the following section.

Line 217. The heading of this section is ambiguous. A correlation between ARGs and phenotypic results would be valuable, at least for the top prevalent STs. Besides, a deeper analysis of fluoroquinolone resistance, the genetic elements and STs might highlight interesting findings

(here, >70% showed resistance to CIP). Disseminated MDR clones seem to be associated with fluoroquinolone resistance (FQR), and many are also producers of CTX-M enzymes, which seriously narrows the treatment options. The WHO and other international institutions also report the worrisome prevalence of FQR among E. coli and global dissemination of FQR determinants within environmental, commensal, and pathogenic organisms (WHO, 2014; ECDC, 2022).

-Line 379. Regarding mcr-1 gene, it is outstanding its presence in 116 out of 499 genomes. In addition, mcr-3.1 was found in four genomes (including 2 mcr-1.1 carriers), and mcr-3.5 in another genome. All these findings should be mentioned.

-Discussion. This section should be strengthened in some places make the most of the enormous amount of data collected.

-Line 447. ST131: there are two isolates of this pandemic clone in the present collection (A113, A304) which are ESBL and FQR. It should be discussed. The same with ST648-F determined in four, including 2 ESBL FQR (A138, A192, A71, A91). (10.3389/fmicb.2018.02659).

-Lines 467-469. I don't agree with this explanation (isolated diseased animals that may have received antibiotic treatment = selection of such resistances). I rather believe that a sustainable administration has caused a strong co selection of MDR successful mechanisms and clones (which is the second part of this paragraph). So, I suggest rewrite this paragraph in favor of a more probable hypothesis.

-Line 475. Please, include the information of the year when China has implemented this ban on the use of antibiotics in feed additives (when this rule became in force).

-Lines 487-488, 496-501 are repetitive.

-Lines 508-516. Here, some discussion on the current prevalence of colistine resistance linked to mcr is lacking. See also comment on Line 379. The phenotypic expression of colistine resistance was not investigated within the collection, was it?

-Methods. Please, visit comment on lines 100-104.

Line 539. The reference 29 is from 2012. ExPEC isolation and identification methods should be mentioned, albeit briefly.

-Line 563. "Mostgenomic analysis". please specify "most".

-References. Reference 76: the journal is missing.

5. Is the methodology sound? Does the work meet the expected standards in your field? Is there enough detail provided in the methods for the work to be reproduced?

The methodology is sound and meets the expected standards in the field. In fact, the results are based on classical lab characterization, as well as omics based on Illumina combined with ONT. From my point of view, there is enough detail in this section for the work to be reproducible.

REVIEWER COMMENTS

Reviewer #1 (Remarks to the Author):

Response: We thank the reviewer for the very constructive and thorough reviews that have helped us tremendously to improve this manuscript.

To Whom It May Concern:

While the role of poultry in the occurrence of human extraintestinal pathogenic *E. coli* infections has been explored in many studies, few have examined the link between those from swine. This research by Li et al is well executed, well communicated and is a considerable contribution to the field.

There is a great need to identify emerging pathogens which may threaten human and animal health as well as those which pose risks to industry and the environment. I congratulate the authors on their efforts, the quality of their work and their contribution to the literature. The strains described in this article are concerning, particularly given their known role in human infections (at the ST level) and their alarming resistance profiles.

Generally speaking, the methodology is sound, though future studies carried out by the authors would benefit from refreshed methodology (described later in more detail), and deposition of code in a publicly available repository (i.e. on github) would improve reproducibility.

Response: We have uploaded all custom codes, along with their corresponding input and output files, to GitHub (available at https://github.com/xdli009/Swine_ExPEC_genomes_analysis.git). We also added a README file to describe all the uploaded files.

I appreciate the breadth and quality of the data visualisation. While the authors have clearly reviewed a great deal of literature, some key articles detailing established links between swine and human sourced *E. coli*, and other relevant findings, were not referenced by the authors. In general, conclusions drawn by the author are well

substantiated, however I have included below some relevant commentary and specific papers which the authors seem to have missed during their research which I think can improve their article.

Response: Thanking the reviewer for providing the relevant papers, which we have used to make revisions as suggested (see below).

This manuscript provides compelling evidence that the relentless use of first-generation antibiotics has led to the evolution of genetic platforms that remain at the core of complex resistance regions that are found in these invasive ExPEC populations with genes encoding resistance to clinically important antibiotics being added (but not replacing) to these platforms. This is a worrying trend and a true One Health threat.

I will include more commentary below.

Kind regards,

Dr Max Cummins

More thoughts:

- Title: perhaps Diversity “and” population structure is more precise

Response: We have revised the title as follows: “Population structure and antibiotic resistance of swine extraintestinal pathogenic *Escherichia coli* from China”

- Aim on line 56 is very broad and beyond the scope of the current manuscript in its current form.

Response: We have revised this sentence as follows: “To explore the diversity and transmission of ExPEC”. This content is in line 62.

- AbuOun et al, 2020 (<https://doi.org/10.3389/fmicb.2020.00861>) details (in Table 7) that many of the same STs identified in the present article are also observed in the gut of healthy pigs. This might indicate that the ExPEC in these swine have a gastrointestinal origin.

Response: ExPEC, as facultative pathogens, are a component of the normal gut flora,

coexisting as commensals (<https://doi.org/10.1016/j.ijmm.2011.09.006>). Occasionally, they can inhabit extraintestinal environments and precipitate severe illnesses. So we agree with the reviewer's viewpoint and added this information into the revised manuscript, with the sentence as follows: “Moreover, several prevalent STs identified in our dataset coincide with those observed in healthy pig intestinal *E. coli*, such as ST10, ST58, ST88, ST101, ST117, and ST744. These findings further support the inference that ExPEC originates from fecal sources and exists asymptotically within the intestinal tract.” This content, along with its corresponding references, is in lines 427-430.

- High carriage rates of B1 and A phylogroup *E. coli* in the swine gut has been reported, which is unsurprising given commercial pig breeds are globally distributed and diets are largely consistent across major swine producing countries (although AMR stewardship practices vary). The discussion would benefit from more detail in this context. The following papers detail this:

- o Bonvegna et al, 2022; <https://doi.org/10.3390/antibiotics11121774>

- o Zingali et al, 2020; <https://doi.org/10.3390/microorganisms8060843>

- o Reid et al, 2017; <https://doi.org/10.1099/mgen.0.000143>

Response: The *E. coli* analyzed in these studies were all isolated from the feces of healthy pigs. Hence, we can integrate this information with the preceding statement for further elaboration in the revised manuscript, with the sentence as follows: “Our large-scale population genomic analysis revealed that more than 80% of swine ExPEC belonged to phylogroups B1 and A. In pig farms worldwide, encompassing regions such as Australia and Poland, the population structure of *E. coli* isolated from the feces of healthy pigs was also predominantly characterized by phylogroups B1 and A. Moreover, several prevalent STs identified in our dataset coincide with those observed in healthy pig intestinal *E. coli*, such as ST10, ST58, ST88, ST101, ST117, and ST744. These findings further support the inference that ExPEC originates from fecal sources and exists asymptotically within the intestinal tract.” This content, along with its corresponding references, is in lines 423-430.

- Authors do not appear to comment on the pathogenesis of these lung infections. Are

pigs inhaling faecal dust (as is common in poultry operations)? This is important context for assessing the risk posed by these pathogens to human and animal health

Response: We have previously investigated the pathogenicity of ExPEC in pigs (<https://doi.org/10.1186/s12864-015-1890-9>). The results showed that infection in healthy pigs can lead to symptoms such as lying on the side, abdominal breathing, shaking, convulsions, lameness, etc. Upon dissection, severe hemorrhagic changes in the lungs were also observed. However, there is limited research on the pathogenic origins of lung infections, such as pig inhalation of fecal dust. Nonetheless, we agree with the reviewer's point; as ExPEC originates from the intestinal tract, there is some correlation with pigs inhaling fecal particles leading to lung infections. Our discussion also elaborated on the relevance of ExPEC originating from feces.

- I encourage the authors to provide data on the carriage of class 1 integrase gene (*intI1*), as these loci play critical roles in promoting multidrug resistance and in tandem with IS26 elements. These *intI1* genes are often truncated by IS26; their detection will probably require flexible coverage cut-offs. If the role of class 1 integrons in the evolution and dissemination of multidrug resistance is not clear to the authors, please refer to Gillings et al (<https://doi.org/10.1128/JB.00152-08>).

Response: We accepted the suggestion and used the tool IntegronFinder 2.0 (https://github.com/gem-pasteur/Integron_Finder) to identify all integrons within 20 complete genomes. The result regarding the association with IS26 elements was also observed. We have included information about integrons in the revised manuscript, with the sentence as follows: “Additionally, we also identified integrons within the 20 complete genomes. A total of 16 integrons were detected, all of which were located within the plasmid genomes (Supplementary Data 8). Furthermore, we found that these integrons generally contain two ARG cassettes: *aadA2/aadA5* and *dfrA12/dfrA17*. It is noteworthy that we observed these integrons closely associated with IS26 (Fig. 4a). This suggests that the accumulation and dissemination of specific ARGs in swine ExPEC were facilitated by the interplay between integrons and transposons.” This content is in lines 329-336.

- While this paper is already bursting at the seams with quality data, large scale SNP

analyses would be useful to determine whether isolates closely related to those under study exist in swine elsewhere, or are associated with other environments and hosts. cgMLST-based approaches (<https://github.com/MDU-PHL/Coreugate>), or highly scalable SNP-analyses like SKA (<https://github.com/simonrharris/SKA>) or SKA2 (<https://github.com/bacpop/ska.rust>) would allow the authors to potentially find evidence of clonally related isolates in public databases (e.g. <https://enterobase.warwick.ac.uk/>), from which members of the primary STs identified could be downloaded and compared with those from the present study.

Response: We accepted the suggestion. We compared swine ExPEC strains with all *E. coli* strains present in the Enterobase database using core genome multi-locus sequence typing (cgMLST). We performed cgMLST analysis for each strain using cgMLSTFinder v.1.2.0 (<https://bitbucket.org/genomicepidemiology/cgmlstfinder>) with the Enterobase *Escherichia/Shigella* cgMLST v1 scheme. This tool was developed by the Center for Genomic Epidemiology (<https://cge.food.dtu.dk/services/cgMLSTFinder/>). The results provided the closest cgST type along with the number of matching alleles for each strain. Based on this, we further filtered the results with a threshold of ≤ 10 allele differences compared to the closest cgST type. Consequently, in the filtered results, the ExPEC strains assigned with cgST types shared a common clonal origin with strains in the Enterobase database belonging to the same cgST types. Information regarding the source of public isolates (e.g., bovine, human) was extracted from the "Source Type" and "Source Details" columns in Enterobase. This method was added in lines 616-626.

We identified 39 cases of genetic relatedness between swine ExPEC and isolates collected from different sources, with a maximum separation of 10 alleles (Supplementary Data 2). Among these occurrences, 21 instances involved *E. coli* strains from the Enterobase database, with explicitly defined host types, including human, swine, and bovine. Human-host instances were the most frequently observed, with a total of 15 swine ExPEC isolates showing genetic relatedness to *E. coli* from various human body sites, such as blood and rectum. Among these, two strains belonging to ST648 (A71 and A91) and one human blood infection strain (Enterobase ID,

ESC_LA6470AA) differed by only 2 alleles, indicating very closely related genomes. Moreover, swine-derived ExPEC belonging to ST410, ST101, ST131, ST167, or ST744 were also observed to share a common clonal origin with human-derived *E. coli*. This result underscores evidence of the zoonotic potential of ExPEC strains originating from swine. It is an important result, and we have added this information in lines 143-157.

Moreover, 13 out of 15 strains with zoonotic potential also contained ESBL genes. we have added this information in lines 255-256. Among 20 isolates with complete genomes, strains A71 and A251 possess zoonotic potential. Plasmid A251_p0 is homologous to plasmid pMCR-H8 from ESBL-producing *E. coli* H8, isolated from the mink farmer in Shandong Province of China. This plasmid harbors a diverse array of ARGs conferring resistance to crucially important antibiotics for human treatment, including *mcr-1* (colistin), *bla_{CTX-M-14}* (cephalosporin), *floR* (florfenicol), *fosA3* (fosfomycin), and *oqxA* and *oqxB* (quinolone). The content is in lines 394-399, and the description of the zoonotic potential of the A251 strain has been added based on this.

Correspondingly, we added relevant content in the discussion, with the sentences as follows: “The most prevalent STs of human ExPEC, such as ST131, ST69, ST95, ST648, ST73, ST393, and ST12, were not commonly found in our dataset. However, a small proportion of swine ExPEC belonged to these ST types, including ST131, ST69, ST95, and ST648. These strains carried ESBL genes, and the majority of them also harbored genetic determinants of fluoroquinolone resistance. And a study has indicated that swine *E. coli* belonging to ST131 posed a potential zoonotic risk. In our investigation, swine ExPEC belonging to ST131 and ST648 were also found to have zoonotic potential.” This content, along with its corresponding references, is in lines 450-457. Elsewhere, we also included relevant content in the discussion, which can be found in lines 469-471.

Formatting/typographical/technical/linguistic errors:

- “Diverse population structure among swine ExPEC” seems imprecise/incorrect. Perhaps something like “Swine ExPEC under analysis were phylogenetically diverse” or “exhibited a complex phylogenetic structure”

Response: Corrected as suggested

- IS elements should be partially italicised (e.g. IS26), not fully italicised (e.g. IS26 [incorrect]), nor, as the authors have written, in plain format (IS26 [also incorrect])

Response: We revised the formatting of all IS elements, partially italicizing them.

- Line 131 – Was this statistically validated? If not, please choose different language

Response: We did not conduct statistical validation at this stage. Considering this, we have revised the sentences as follows: “We also inspected the isolated organs and times of all the *E. coli* on the basis of phylogenetic analysis. Our findings revealed that strains from special organ or time point were distributed throughout the whole phylogenetic tree, suggesting an absence of association between the source or time of isolation and the phylogroup or ST.” This content is in lines 133-137.

Methodological overview:

Generally speaking, the choice of methodology seems appropriate and well considered, however some of the tools utilised are quite dated and some details are missing relating to parameters utilised for each tool.

Response: We have added parameters for some tools where details were previously missing, to ensure reproducibility of all analyses. Regarding the issue of outdated tools, we have discussed it below.

Perhaps the most consequential methodological choices are those relating to long-read assembly. The methods chosen in this regard are relatively standardised, however this does not mean that they are necessarily appropriate. The authors may also consider stating why hybrid assembly approaches were not taken given that both short-read and long-read sequence data was available.

Response: We fully agree with the reviewer's viewpoint. Indeed, it is crucial to consider both long-read assembly and hybrid assembly to determine which yields superior results. In this study, we initially performed long-read assembly on 20 representative strains. The results revealed that, except for one plasmid in strain A376 which did not circularize, all chromosomes and plasmids circularized successfully. Therefore, owing to the high quality of this assembly result, we directly opted for long-read assembly. Subsequently, the resulting genome sequences underwent two rounds of polishing with

Pilon v1.23, using Illumina reads.

It is also not clear what quality control procedures were utilised and whether or not there were a sufficient number of reads to perform long-read only assemblies; “Their length and quality were assessed” (Line 589) – to what degree?

Response: We employed the tools Guppy and Filtrong for quality control. Specifically, based on the signal data contained in the output FAST5 files generated by Nanopore sequencers, we initially performed basecalling using Guppy v5.0.16 (https://community.nanoporetech.com/docs/prepare/library_prep_protocols/Guppy-protocol/v/gpb_2003_v1_revax_14dec2018/guppy-software-overview) and selectively retained reads that met Guppy’s quality criteria. Guppy, supported by ONT, recommends retaining reads with a mean q-score value above 7. Following this, we applied more stringent filtering measures (utilizing Filtrong v0.2.0) to remove the worst 5% of long-reads.

Finally, we utilized NanoPlot v1.29.1 to calculate various statistics of filtered long-reads, including their length and quality. The results indicated that the mean read quality of these samples was around 9, and the mean read length was around 20 kbp (Supplementary Data 10). Additionally, the mean size of each sample yielding filtered data reached 1168 Mbp, with a minimum of 669 Mbp. The sequencing depth was at least 100x, which was sufficient for each strain to perform long-read only assembly in this study. We have revised the sentences to clarify this process in lines 629-639.

Other general methodological comments

- More information needed on the sections starting line 606: “First, 610 the presence of all ARGs associated with that genomic island must be confirmed in the 611 genome”- Confirmed using which tools?

Response: Here, we utilized a custom Python script to confirm the presence of all ARGs associated with the genomic island. The relevant code has been uploaded to GitHub (https://github.com/xkli009/Swine_ExPEC_genomes_analysis/tree/main/Detection_Genomic_Islands). We have added this information into the revised manuscript.

- Versions of R packages utilised should be included

Response: We have added the version information of all R packages in the revised manuscript.

- The code utilised for data visualisation (i.e. in R) should be provided on github. This is a critical part of maintaining reproducibility and promoting transparency in bioinformatic analyses

Response: We have uploaded all custom scripts, including R codes, to GitHub (https://github.com/xkli009/Swine_ExPEC_genomes_analysis/tree/main/Visualization).

- Line 631 – Based on “jaccard indices” – is this to say that jaccard distance based clustering approaches were utilised? Please clarify language here

Response: Here, the input data for pheatmap is a Jaccard similarity matrix for plasmid genomes. The pairwise Jaccard indices were calculated using the sourmash tool (described in the "Annotation of complete genomes and comparative analysis of plasmids" section of the Methods). Subsequently, pheatmap is used solely for visualization based on this matrix, with `cluster_row` and `cluster_col` set to `False`. The reason for not needing clustering using pheatmap is that the row and column orders of the input matrix have already been adjusted based on Jaccard indices and the composition spectrum of ARGs. Moreover, the corresponding code has been uploaded to GitHub. We have also revised the sentence to clarify this process.

- I would recommend Bakta (<https://github.com/oschwengers/bakta>) be utilised for future investigations instead. Prokka was released 10 years ago has not been updated for five years (<https://github.com/tseemann/prokka/releases>).

A comparison of Prokka, Bakta and other tools can be found here: <https://www.microbiologyresearch.org/content/journal/mgen/10.1099/mgen.0.000685>.

- Similarly, Panaroo (<https://github.com/gtonkinhill/panaroo>) (Tonkin-Hill et al, 2020), rather than Roary, might be a wise choice for future analyses (the former released in the last few years and the latter also released around a decade ago with no updates released since 2019)

A comparison of Panaroo, Roary and other tools can be found here: <https://genomebiology.biomedcentral.com/articles/10.1186/s13059-020-02090-4>

- The authors may also be interested in abritAMR, an ISO-accredited resistance detection tool (<https://github.com/MDU-PHL/abritamr>) (Sherry et al, 2023), which would provide more robust resistance data than that provided by abricate.

- Pilon has been shown to introduce deletion errors by Wick and Holt and I therefore feel its use should be cautioned: (<https://www.biorxiv.org/content/10.1101/2021.10.14.464465v1>). Wick is well regarded for his expertise in long read assembly – if the authors are not familiar with these pages I encourage them to review them and integrate these learnings into their future long-read assembly endeavours. (<https://github.com/rrwick/Tricycler/wiki>; <https://github.com/rrwick/Tricycler/wiki/Guide-to-bacterial-genome-assembly>).

Response: Thank the reviewer for thorough review and insightful suggestions for improving the methodology in our study. We agree with the reviewer's suggestion to employ these refreshed tools in future research. In this study, although some of the tools we utilized were released quite some time ago, they are still reasonable and commonly used. For instance, the tool Prokka has been cited over three thousand times since 2023, while Roary has been cited over a thousand times during the same period. On the other hand, the principle behind the annotation by the Abricate tool is based on the classical Blast method. With appropriate settings for the IDENTITY and COVERAGE thresholds, we believe the annotation results in this study are reliable. Pilon may potentially introduce large deletion errors in regions with repetitive sequences, as error correction of repetitive sequences in the genome using short-read data poses significant challenges. Fortunately, we examined all the logs from Pilon runs and found that indels (insertions and deletions) corrected by Pilon were all small, with no instances of large indels identified. Therefore, we consider the genome assembly results in this study to be reliable. In summary, this study continues to employ the previous methods. However, we have also taken note of the refreshed tools recommended by the reviewer, and we intend to incorporate them into our future research endeavors.

Reviewer #2 (Remarks to the Author):

Response: We express our gratitude to the reviewer for the constructive feedback, which has greatly contributed to the enhancement of this manuscript.

The paper reports on a comprehensive investigation of 499 *E. coli* isolates from clinical lesions of pigs from different locations in China. Unfortunately there is no information about how *E. coli* could be linked to the observed lesions including presence of other microbiological agents and it is difficult to designate the isolates as ExPEC on this background.

Response: We apologize for the confusion caused by the brief mention in the previous manuscript regarding the identification of ExPEC, which only stated its consistency with our previous research methods (<https://doi.org/10.1016/j.tvjl.2011.06.038>) without further elaboration. In our study, pigs from which we isolate *E. coli* exhibit extraintestinal infections, such as septicaemia, meningitis, and respiratory diseases. Each *E. coli* from the tissue samples of pigs was confirmed using a rapid identification system (Biolog MicroStation). Furthermore, the method for identifying ExPEC involves multiplex PCR utilizing specific oligonucleotide primers to amplify five virulence genes: *papA* and/or *papC* (P fimbriae), *sfa/foc* (S and F1C fimbriae), *afa/dra* (Dr-antigen-binding adhesins), *iutA* (aerobactin receptor), and *kpsMT II* (group 2 capsular polysaccharide units). ExPEC was characterized by the presence of at least two of the aforementioned five markers (10.1128/AAC.47.7.2161-2168.2003). We have added this information in lines 562-566.

Antimicrobial susceptibility was analysed. The genomes of the isolates were generated and analysed according to best standards including predictions of phylogroup, MLST, virulence genes and antimicrobial resistance genes. A high frequency of multi resistant isolates were predicted with a link to ST410. Based on closed genomes the genetic background for a selection of antibiotic resistance - and potential virulence genes was analysed. Results are discussed in relation to similar *E. coli* populations obtained from humans and some overlaps were found with respect to sequence types and virulence profiles. Unfortunately there is no comparison to *E. coli* isolated from diarrhea of weaned pigs which is the reason for most use of antibiotics. It would have been

interesting to know the proportion of ExPEC types isolated from diarrhea and relate the problem to *E. coli* isolated from other lesion types. It would also have been interesting to compare the results to *E. coli* isolated from diarrhea of pigs.

Response: We agree with the reviewer's suggestion that comparing the *E. coli* isolated from diarrhea of weaned pigs is highly meaningful. Postweaning diarrhea (PWD) is often triggered by infections from enterotoxigenic *E. coli* (ETEC), mainly including F4 (K88)⁺ and F18⁺ *E. coli* (<https://doi.org/10.1016/j.aninu.2017.10.001>). Therefore, to confirmed the proportion of ExPEC types associated with diarrhea, we further identified operons encoding fimbriae (F4, F5, F6, F18, and F41) in 499 ExPEC genomes. Combining these findings with our previous results on the identification of virulence factors associated with enteric diseases, we found that 49 strains harbored one or more of the following: the Shiga-toxin (STX), heat-labile (LT) enterotoxin, heat-stable (ST) enterotoxin, intimin, and fimbrial adhesins (F4, F5, F18, and F41). These strains mainly belonged to phylogroups F, A, and B1. It is noteworthy that these strains may represent hybrid pathotypes. Furthermore, based on these virulence factors, we classified these 49 strains as four hybrid pathotypes, including ExPEC/ETEC, ExPEC/EPEC, ExPEC/EHEC, and ExPEC/EHEC/ETEC (Supplementary Data 3). The most prevalent hybrid pathotype is ExPEC/ETEC, comprising 39 strains in total. Roughly 70% of these strains contain the F5 fimbriae, all of which belonged to ST354/ST457. We have added this information in lines 166-174.

Correspondingly, the results were discussed in relation to the overlap of certain sequence types between swine ExPEC and ETEC isolates causing PWD, with the sentences as follows: “Although only a small number of ExPEC strains contain toxins typical of enteric *E. coli* pathotypes, caution must be exercised regarding these potential hybrid pathotypic strains. For instance, ETEC strains, mainly including F4⁺ and F18⁺ *E. coli*, were often the cause of postweaning diarrhea (PWD), which incurs significant economic costs for the pig industry. F4-positive ETEC isolates causing PWD commonly belonged to ST100, while F18-positive isolates were assigned to diverse STs, including ST23, ST10, ST155. In our study, all three F4-positive ExPEC/ETEC strains belonged to ST100, and one F18-positive ExPEC/ETEC strain belonged to

ST155. This caution arises from the potential of hybrid pathotypic *E. coli* to cause more severe illnesses and exhibit higher levels of resistance compared to non-hybrid pathotypes. Furthermore, our investigation illustrated that swine-derived hybrid pathotypic strains displayed heightened antibiotic resistance.” This content, along with its corresponding references, is in lines 435-447.

Reviewer #3 (Remarks to the Author):

Response: We greatly appreciate and thank the reviewer for providing us with the precious comments on this work which have helped us greatly improve the manuscript.

1. What are the noteworthy results?

The present study shows a comprehensive picture of the putative contribution of swine derived ExPEC to human infections, based on large-scale genome sequencing and antibiotic resistance tests of 499 isolates (central China, 2011-2017).

It is noteworthy the huge genomic diversity found within this swine ExPEC collection in terms of phylogeny, serotypes (200 O:H combinations), genetic virulence or AMR. Important high-risk clones were determined, such as ST23, ST88, ST117, ST131, ST648. It is also of note that >50% of the collection were ESBL producers and >70% showed FQR. To highlight the 406 combinations of different ARGs found among the 492 genomes containing at least 2 ARGs, with an average of 13 ARGs. The top prevalent ARGs were all frequently found in both chromosome and plasmid genomes, such as *sul1*, *sul2*, *floR*, *tet(A)*, *blaTEM-1*, *strA*, *strB*. The authors found some alarming cooccurrences targeting human and veterinary drugs, such as *blaCTX-M-55*, *blaCTX-M-14*, *fosA3*, and *mcr-1*. All *fosA3* genes and ESBL genes were associated with IS26. Notably, among the three plasmids coexisting with *mcr-1*, the gene *blaCTX-M-14*, and *fosA* constitute a potential co-transfer unit with two IS26 elements serving as flanking sequences. Moreover, two of the four swine isolates that showed resistance to carbapenems were selected for sequencing of closed genomes; as a result, two plasmids belonging to IncX3 (A382_p1) and IncFIA/FIC (A376_p2) harbored 362 *blaNDM-1*

and blaNDM-5, respectively.

2. Will the work be of significance to the field and related fields? How does it compare to the established literature? If the work is not original, please provide relevant references.

The present work is of significance to the field exploring the connection of animal ExPEC to human infections. It is a comprehensive analysis of multidrug resistant mechanisms that reveals potential transmission pathways of ARGs within swine ExPEC and to human pathogens. It thus provides information on a topic that has been little explored to date, and where the control of ExPEC infections is facing increasing complication by the emergence of ARGs.

3. Does the work support the conclusions and claims, or is additional evidence needed? The conclusions are consistent with the results obtained here. There is no need for further evidence. It is suggested, however, to reinforce some aspects by making some missing information relevant (see explanation below).

Response: Thanking the reviewer for providing the relevant information, which we have used to make revisions as suggested (see below).

4. Are there any flaws in the data analysis, interpretation and conclusions? Do these prohibit publication or require revision?

There are no flaws in the data analysis, interpretation, and conclusions. However, some additional information and discussion would enrich the results presented here; specifically on the points listed below. So, my suggestion is moderate revision.

-Abstract. Some general phrases could be removed or simplified (e.g. the two first could be "The diversity and antibiotic resistance of animal extraintestinal pathogenic *Escherichia coli* (ExPEC) and its connection to human infections, remain largely unexplored.") in favor of more specific information on ARG co-occurrence or the suggested transmission pathways based on the plasmid analysis. Also some information on the VF should be mentioned.

Response: We have enhanced the abstract by including detailed information on transmission pathways, with the sentence as follows: "Furthermore, assembly of 20 complete genomes illuminated transmission pathways of ARGs within swine ExPEC

and to human pathogens. Particularly noteworthy were ARGs relevant to both human and veterinary medicine, such as transmission of plasmid co-harboring *fosA3*, *bla*_{CTX-M-14}, and *mcr-1* genes between swine ExPEC and human-origin *Salmonella enterica*.”

We also added the information about VF to the abstract, with the sentence as follows: “Additionally, 49 ExPEC strains harbored toxins typical of enteric *E. coli* pathotypes, implying hybrid pathotypes like ExPEC/ETEC and ExPEC/EPEC.”

-Introduction. Line 75-76. It would be interesting to add here information on the role of food-producing animals as a potential source of human ExPEC (including UPEC), as it has been recently proved by different authors including high-risk clones such as ST69, ST95, ST117, ST648 or ST131 (Riley, 2020 [10.1146/annurev-food-032519-051618](https://doi.org/10.1146/annurev-food-032519-051618); Díaz-Jiménez et al., 2021 <https://doi.org/10.1016/j.foodcont.2020.107713>).

Response: We added this information into the revised manuscript, with the sentence as follows: “It's noteworthy that foodborne *E. coli* can also cause extraintestinal infections in humans, involving high-risk clones such as ST69, ST95, ST117, ST131, and ST648.” This content, along with its corresponding references, is in lines 85-87.

Lines 83-85. While this statement is true regarding the in-depth analysis of ARGs, there are some distinct approaches comparing animal ExPEC and those of human infections using different in vitro and in vivo assays that might complete this introduction and/or the discussion (e.g. Zhuge et al. 2020 <https://doi.org/10.1111/tbed.13755>; García et al. 2023 <https://doi.org/10.1016/j.onehlt.2023.100558>).

Response: We added this information into the revised manuscript, with the sentences as follows: “Many studies have found that *E. coli* from animal reservoirs may be a potential source for human ExPEC, especially those from poultry. Moreover, evidence of the pathogenicity of poultry-derived *E. coli* strains in causing extraintestinal infections in humans has been demonstrated through *in vivo* and *in vitro* experiments.” This content, along with its corresponding references, is in lines 81-85.

-Results. Lines 100-104. Here or in Methods: Even if some of this information is partially shown in the Supplementary Data 1, it would be interesting to include data on how many farms, how many geographic regions, and whether isolates are unique or duplicates per animal (outbreaks? isolated cases? Age of the animals/production stage?)

to reinforce the idea of the representativeness of the ExPEC collection studied here.

Response: Due to production privacy of the pig farms, we did not collect detailed information on disease outbreaks and the stages of pig breeding. Nonetheless, the pig farms from which we isolate ExPEC strains can be geographically located by province. Specifically, 499 *E. coli* strains were collected from 23 Chinese provinces, mainly from Hubei, Henan and Hunan provinces. Furthermore, we only isolated and preserved one representative ExPEC strain from each pig farm at a given time, ensuring that each strain originates from distinct pigs. We have added this information in lines 111-116.

Line 114. Rewrite “.. 490 of 499 *E. coli* could be assigned to 75 STs...”

Response: Corrected as suggested.

Lines 121-123. This phrase would belong to the discussion section in any case; however, I do not think it is an accurate affirmation without more data because it depends on the geographical area, production status...

Response: We deleted this sentence.

Line 134. I suggest removing this line “we also found no certain serotype associated with special isolated time or organ.”

Response: Corrected as suggested.

80% of the *E. coli* were isolated from lung (I am curious about this: was *E. coli* as the only pathogen present? Was it found in other organs of the animal? Was *E. coli* linked to a respiratory pathology?). “diseased tissues” is the definition of the origin of the isolates (line 100) but there is no mention of the pathologies.

Response: In current clinical practice with pigs, ExPEC can co-infect with other bacteria such as *Streptococcus*. This study primarily focused on ExPEC research, so there is no statistical data available on the isolation and identification of other bacteria. In our previous research, it was observed that following infection with highly virulent ExPEC strains, pigs exhibited dissemination of the bacteria across multiple organs or anatomical sites, often manifesting in respiratory symptoms. In addition, we infected mice with multiple isolates of ExPEC strains obtained from pig lungs, nearly half of which exhibited high pathogenicity (<https://doi.org/10.1016/j.tvjl.2011.06.038>).

In our study, pigs from which we isolate *E. coli* exhibit extraintestinal infections, such as septicaemia, meningitis, and respiratory diseases. Identification of ExPEC is

achieved by amplifying representative virulence gene markers. We have added this information in the Methods section.

Lines 141-167. I wonder if the authors searched fimbriae associated with ETEC and or STEC pathotypes such as F4, F5, F6, F18, and F41 in the studied collection. If so, please add. According to Supplementary Figure 3, some isolates exhibit hybrid pathotypes (ETEC/STEC/ExPEC, EPEC/ExPEC). This finding is important, even if they are only a few (line 149): to which STs do they belong? Are they ESBL/MDR?

Response: We have searched for fimbriae associated with ETEC and STEC pathotypes as suggested. (Supplementary Fig. 3 and Supplementary Data 3). Our method involved the identification of operons encoding fimbriae, such as Fan operon which expresses F5 fimbriae. Combining our previous results on toxin genes associated with enteric diseases, we found that 49 strains harbored one or more of the following: the Shiga-toxin (STX), heat-labile (LT) enterotoxin, heat-stable (ST) enterotoxin, intimin, and fimbrial adhesins (F4, F5, F18, and F41). These strains mainly belonged to phylogroups F, A, and B1. Furthermore, based on these virulence factors, we classified these 49 strains as four hybrid pathotypes, including ExPEC/ETEC, ExPEC/EPEC, ExPEC/EHEC, and ExPEC/EHEC/ETEC. The most prevalent hybrid pathotype is ExPEC/ETEC, comprising 39 strains in total. Roughly 70% of these strains contain the F5 fimbriae, all of which belonged to ST354/ST457. We have added this information in lines 166-174.

Among these hybrid pathotypic strains, 46 were MDR, of which 54.3% displayed resistance to agents across all 7 classes. It is an important result, and we have added this information in lines 206-208. When focusing on all ESBL genes present in our dataset, we found that up to 262 strains harbored at least one ESBL gene. Moreover, among these, 25 strains were identified as hybrid pathotypes, specifically ExPEC/ETEC and ExPEC/EPEC (Supplementary Data 6). We have added this information in lines 252-255.

Correspondingly, we also added relevant content in the discussion, with the sentences as follows: “Although only a small number of ExPEC strains contain toxins typical of enteric *E. coli* pathotypes, caution must be exercised regarding these potential

hybrid pathotypic strains. For instance, ETEC strains, mainly including F4⁺ and F18⁺ *E. coli*, were often the cause of postweaning diarrhea (PWD), which incurs significant economic costs for the pig industry. F4-positive ETEC isolates causing PWD commonly belonged to ST100, while F18-positive isolates were assigned to diverse STs, including ST23, ST10, ST155. In our study, all three F4-positive ExPEC/ETEC strains belonged to ST100, and one F18-positive ExPEC/ETEC strain belonged to ST155. This caution arises from the potential of hybrid pathotypic *E. coli* to cause more severe illnesses and exhibit higher levels of resistance compared to non-hybrid pathotypes. Furthermore, our investigation illustrated that swine-derived hybrid pathotypic strains displayed heightened antibiotic resistance.” This content, along with its corresponding references, is in lines 435-447.

Line 169 “In the past decades, antibiotics were heavily used in poultry and livestock industries” where? in China?. Please, complete.

Response: Yes, in China. We added this information into the revised manuscript.

Line 171. Systematically?

Response: We deleted this word.

Line 175. What do the authors mean with “unambiguous results”. Probably the meaning is: MDR isolates were defined following the Magiorakos et al. (2012) criteria, as those showing acquired non-susceptibility to at least one agent in three or more antimicrobial categories. Please, include the following reference for the definition of MDR: Magiorakos et al. Clin. Microbiol. Infect., 18 (2012), pp. 268-281, 10.1111/j.1469-0691.2011.03570.x

Response: Each time an antibiotic susceptibility test was conducted, control strains were used. If the susceptibility results of the control strains do not match the actual outcomes, it indicates that the results are unreliable. Therefore, 'unambiguous results' denote having clear and dependable antibiotic susceptibility outcomes. We have changed “unambiguous results” to “dependable antibiotic susceptibility outcomes”. The description of using control strains in antibiotic susceptibility tests had been outlined in the Methods section. In addition, we added this reference for the definition of MDR in the revised manuscript.

Lines 178-181. “strains A376 and A430 exhibiting resistance to all 18 antibiotics tested”: 18 out of the 20 antibiotics tested?. The last part of this paragraph should be removed or at least discussed in detail in the following section.

Response: Yes, 18 out of the 20 antibiotics tested. We revised this to “strains A376 and A430 exhibiting resistance to 18 antibiotics tested”. In addition, we deleted the last part of this paragraph.

Lines 201-202. The same for the following phrase: “These findings provide a plausible explanation for the popularity of ST410 as the predominant sequence type”. There is no consistency in the argument to confirm this. So, I suggest that this should be removed or discuss in detail in the following section.

Response: We deleted this sentence.

Line 217. The heading of this section is ambiguous. A correlation between ARGs and phenotypic results would be valuable, at least for the top prevalent STs. Besides, a deeper analysis of fluoroquinolone resistance, the genetic elements and STs might highlight interesting findings (here, >70% showed resistance to CIP). Disseminated MDR clones seem to be associated with fluoroquinolone resistance (FQR), and many are also producers of CTX-M enzymes, which seriously narrows the treatment options. The WHO and other international institutions also report the worrisome prevalence of FQR among *E. coli* and global dissemination of FQR determinants within environmental, commensal, and pathogenic organisms (WHO, 2014; ECDC, 2022).

Response: We revised this heading to “Swine ExPEC genomes harbor diverse and abundant antibiotic resistance genes”. Furthermore, we conducted a deeper analysis of FQR, the genetic elements, and STs. Specifically, in the most prevalent ST (ST410), we explored the correlation between FQR phenotypes and genetic determinants. Firstly, based on the comparative results of antibiotic resistance phenotypes among the four top prevalent STs, we noted that strains belonging to ST410 exhibited the highest resistance (close to 100%) to three fluoroquinolones (Fig 2c). Therefore, we analyzed the distribution of genetic determinants responsible for fluoroquinolone resistance, including ARGs and chromosomal point mutations, within the ExPEC population (Supplementary Fig. 5). We observed mutations in three chromosomal genes (*gyrA*,

parC, and *parE*) present in all ST410 strains, and the prevalence of fluoroquinolone resistance genes in ST410 was higher compared to the other prevalent STs. This could be one of the important genetic factors contributing to the prevalence of ST410. It's worth noting that all ST410 strains were identified as MDR. Moreover, a similar pattern was observed in ST354, which presented the hybrid pathotype ExPEC/ETEC, with most strains belonging to this ST type containing ESBL genes. We have added this result in lines 266-277.

Correspondingly, we also added relevant content in the discussion, with the sentences as follows: "ST354 represented predominant ST of fluoroquinolone-resistant *E. coli* originating from canine fecal and extraintestinal infections, with certain strains within this ST type carrying ESBL genes. In our study, swine ExPEC of ST354 also exhibited similar characteristics: the hybrid pathotype, resistance to fluoroquinolones, and the presence of ESBL genes." This content, along with its corresponding references, is in lines 461-465.

-Line 379. Regarding *mcr-1* gene, it is outstanding its presence in 116 out of 499 genomes. In addition, *mcr-3.1* was found in four genomes (including 2 *mcr-1.1* carriers), and *mcr-3.5* in another genome. All these findings should be mentioned.

Response: We added this information in the revised manuscript, with the sentence as follows: "Additionally, we observed that despite *mcr-3* being detected in only five genomes (*mcr-3.1* in four genomes and *mcr-3.5* in one genome), it is noteworthy that two of these genomes concurrently carry both *mcr-1* and *mcr-3* (Supplementary Data 5)." Due to its relevance to the co-occurrence of ARGs, this sentence was added to the result - "Highly diverse antibiotic resistance gene co-occurrences contribute to the high prevalence of multidrug-resistant (MDR) ExPEC".

-Discussion. This section should be strengthened in some places make the most of the enormous amount of data collected.

-Line 447. ST131: there are two isolates of this pandemic clone in the present collection (A113, A304) which are ESBL and FQR. It should be discussed. The same with ST648-F determined in four, including 2 ESBL FQR (A138, A192, A71, A91). (10.3389/fmicb.2018.02659).

Response: We added this information in the discussion, with the sentence as follows: “The most prevalent STs of human ExPEC, such as ST131, ST69, ST95, ST648, ST73, ST393, and ST12, were not commonly found in our dataset. However, a small proportion of swine ExPEC belonged to these ST types, including ST131, ST69, ST95, and ST648. These strains carried ESBL genes, and the majority of them also harbored genetic determinants of fluoroquinolone resistance. And a study has indicated that swine *E. coli* belonging to ST131 posed a potential zoonotic risk.” This content, along with its corresponding references, is in lines 450-456.

-Lines 467-469. I don't agree with this explanation (isolated diseased animals that may have received antibiotic treatment = selection of such resistances). I rather believe that a sustainable administration has caused a strong co selection of MDR successful mechanisms and clones (which is the second part of this paragraph). So, I suggest rewrite this paragraph in favor of a more probable hypothesis.

Response: We rewrote this part as follows: “Our collection of *E. coli* strains exhibited extensive resistance to various classes of drugs, surpassing the severity of resistance levels reported previously. Particularly noteworthy was the heightened prevalence of MDR within swine ExPEC. This phenomenon may be attributed to the enduring use of antibiotics in China's livestock industry over the preceding decades.” This content, along with its corresponding references, is in lines 479-483.

-Line 475. Please, include the information of the year when China has implemented this ban on the use of antibiotics in feed additives (when this rule became in force).

Response: In 2020, China implemented this ban. We have added this information into the revised manuscript.

-Lines 487-488, 496-501 are repetitive.

Response: We deleted the contents of lines 487-488. The paragraph containing lines 496-501 primarily discusses horizontal gene transfer of ARGs, hence lines 496-501 were retained.

-Lines 508-516. Here, some discussion on the current prevalence of colistine resistance linked to *mcr* is lacking. See also comment on Line 379. The phenotypic expression of colistine resistance was not investigated within the collection, was it?

Response: We have supplemented the relevant content as follows: “ Colistin resistance genes encompass various types, with *mcr-1* and *mcr-3* having been disseminated worldwide. It is notable that strains simultaneously carrying both *mcr-1* and *mcr-3* have been isolated from various sample types, encompassing cattle in Spain and Italy, as well as pigs, poultry, and clinical samples in China. Our study describes two swine ExPEC strains concurrently harboring *mcr-1* and *mcr-3*. While this occurrence is not prevalent, vigilance regarding the risk of its accumulation is warranted.” This content, along with its corresponding references, is in lines 528-534.

We did not investigate the phenotypic expression of colistine resistance.

-Methods. Please, visit comment on lines 100-104.

Response: We have included relevant information as suggested.

Line 539. The reference 29 is from 2012. ExPEC isolation and identification methods should be mentioned, albeit briefly.

Response: We added a brief description in lines 562-566, with the sentence as follows: “In brief, the method for identifying ExPEC involves multiplex PCR utilizing specific oligonucleotide primers to amplify five virulence genes: *papA* and/or *papC* (P fimbriae), *sfa/foc* (S and F1C fimbriae), *afa/dra* (Dr-antigen-binding adhesins), *iutA* (aerobactin receptor), and *kpsMT II* (group 2 capsular polysaccharide units). ExPEC was characterized by the presence of at least two of the aforementioned five markers.”

-Line 563. “Mostgenomic analysis”. please specify "most".

Response: We have outlined the specific analysis, which is detailed on line 590.

-References. Reference 76: the journal is missing.

Response: We have corrected.

5. Is the methodology sound? Does the work meet the expected standards in your field?

Is there enough detail provided in the methods for the work to be reproduced?

The methodology is sound and meets the expected standards in the field. In fact, the results are based on classical lab characterization, as well as omics based on Illumina combined with ONT. From my point of view, there is enough detail in this section for the work to be reproducible.

REVIEWERS' COMMENTS

Reviewer #1 (Remarks to the Author):

Dear Editor,

The manuscript by Li et al is much improved. I commend the authors and I endorse its publication. I believe the following minor changes should be made as they will take only a few minutes.

Thank you to the authors for producing this high-quality article and thank you to the Editor for the opportunity to contribute to the review process.

Kind regards,
Max

Line 32 - "The results indicated that swine ExPEC under analysis was phylogenetically diverse "
This should read 'were' diverse

Line 137 - "Additionally, to understand the epidemiology of these swine ExPEC strains, we compared them with all E. coli strains present in the EnteroBase database using core genome multi-locus sequence typing (cgMLST)."
A count of genomes under analysis from Enterobase and an access date is required

Line 395 - "Importantly, the A251 strain possesses zoonotic potential."
This statement should be substantiated with genotypic data, phenotypic data or a citation

Reviewer #2 (Remarks to the Author):

The introduction is not really considering previous studies of ExPEC in pigs. The reason to perform the study should be provided. For instance, there are publications of sow urinary tract infection with ExPEC:

<https://pubmed.ncbi.nlm.nih.gov/?term=urinary+tract+infections+pig+expec&sort=pubdate>

English needs improvement with respect to grammar and past/present tense.

The introduction is systematically in past tense, however sometimes this causes confusion where published observations can be confused with observation from the current study. Examples can be found in L59, 60, 63.

Oral phrasing is used for instance L85 and other places.

L28, ExPEC are plural since there are many bacteria: pose

L33, again plural: were

L41-42, 20 complete genomes cannot be assembled and therefore analysis instead of assembly or something similar.

L44, plasmids

L51, repeat ExPEC and not "it" and again plural: are.

L112, past tense more appropriate: originated

L135, insert to "a special"

L140, insert strains or isolates instead of members

L146, describe the degree of genetic relatedness in some quantitative term.

L156, E. coli in ital font

L202, "dependable antibiotic susceptibility outcomes" is unclear.

L248, 253, do you count blaTEM-1B as ESBL? it is not normally considered as providing broad spectrum resistance.

L268, "exhibited the highest resistance" compared to other STs?

L285-286, unclear how 340 combinations can be present in a single genome.

L290, "fighting"? "putative MDR isolates"?

L302, maybe plural since they are classes and do not include single antibiotics
L344, located instead of situated.
L563-565 unclear if you tested for combinations *sfa/foc*, *afa/dra*
or these are alternative gene designations.
L577, relied
L579, belonged
L586, ital font for *E. coli*

Reviewer #3 (Remarks to the Author):

In this second version, the authors have considered all the reviewers' suggestions to complete the required information and to perform a deeper analysis of their results. From my point of view, this new version has improved in terms of reinforcing the results and conclusions obtained by their research.

All the issues posed have been satisfactorily addressed, so I recommend the publication of this work in its current state.

Reviewer #1 (Remarks to the Author):

Dear Editor,

The manuscript by Li et al is much improved. I commend the authors and I endorse its publication. I believe the following minor changes should be made as they will take only a few minutes.

Thank you to the authors for producing this high-quality article and thank you to the Editor for the opportunity to contribute to the review process.

Kind regards,

Max

Line 32 – "The results indicated that swine ExPEC under analysis was phylogenetically diverse "

This should read 'were' diverse

Response: Corrected as suggested.

Line 137 - "Additionally, to understand the epidemiology of these swine ExPEC strains, we compared them with all *E. coli* strains present in the Enterobase database using core genome multi-locus sequence typing (cgMLST)."

A count of genomes under analysis from Enterobase and an access date is required

Response: We retrieved the *E. coli* strains from the Enterobase database on April 20, 2024. In our analysis, we included a total of 229,988 genomes. We have added this information in the " cgMLST analysis" section of the Methods.

Line 395 – "Importantly, the A251 strain possesses zoonotic potential."

This statement should be substantiated with genotypic data, phenotypic data or a citation

Response: This statement was substantiated with genotypic data, which was provided in Supplementary Data 2. We have added this information in the revised manuscript.

Reviewer #2 (Remarks to the Author):

The introduction is not really considering previous studies of ExPEC in pigs. The reason to perform the study should be provided. For instance, there are publications of sow urinary tract infection with ExPEC: <https://pubmed.ncbi.nlm.nih.gov/?term=urinary+tract+infections+pig+expec&sort=pubdate>

Response: Thank the reviewer for the valuable feedback. We have supplemented our introduction to include more detailed information on the research progress related to swine ExPEC. And at the end of the paragraph, we also clarified the reason for conducting our study. In the revised introduction, we have added the following paragraph: " Previously, we and other groups found that ExPEC can lead to significant losses in the swine industry. Swine ExPEC strains are recognized as significant disease agents, causing conditions such as UTI, meningitis, pneumonia, arthritis, and septicemia. The population structure of swine ExPEC varied slightly across different datasets, primarily comprising the phylogroups A, B1, and D. Additionally, uropathogenic *E. coli* (UPEC) from sows predominantly belonged to phylogroups B1, D, and E. Furthermore, one study linked the density of industrial hog production to increased UTI rates in nearby human populations, suggesting that intense hog production may elevate UTI incidence in surrounding communities. This observation was particularly concerning in light of the extensive use of antibiotics in the poultry and livestock industries, as antibiotic resistance has been found to be widespread in strains of *E. coli* from animal sources. Despite these insights, a comprehensive understanding of the diversity and resistance determinants of swine ExPEC, as well as their relationships and potential spread to human pathogens, remains poorly documented. "

English needs improvement with respect to grammar and past/present tense.

The introduction is systematically in past tense, however sometimes this causes

confusion where published observations can be confused with observation from the current study. Examples can be found in L59, 60, 63.

Response: We recognize the importance of clearly distinguishing between published observations and observations from the current study. We have revised the introduction to address these issues, ensuring that generally accepted knowledge and observations in the current study are described in the present tense, while past studies are described in the past tense.

Oral phrasing is used for instance L85 and other places.

Response: We have carefully reviewed the manuscript for colloquial expressions and revised them to ensure a more formal and academic tone.

L28, ExPEC are plural since there are many bacteria: pose

Response: Corrected as suggested.

L33, again plural: were

Response: Corrected as suggested.

L41-42, 20 complete genomes cannot be assembled and therefore analysis instead of assembly or something similar.

Response: We have replaced 'assembly' with 'analysis'.

L44, plasmids

Response: Corrected as suggested.

L51, repeat ExPEC and not "it" and again plural: are.

Response: Corrected as suggested.

L112, past tense more appropriate: originated

Response: Corrected as suggested.

L135, insert to "a special"

Response: Corrected as suggested.

L140, insert strains or isolates instead of members

Response: We have replaced 'strains' with 'members'.

L146, describe the degree of genetic relatedness in some quantitative term.

Response: Thank the reviewer for the insightful feedback regarding the description of genetic relatedness. In the original sentence, we specified that the genetic relatedness

was defined by a maximum separation of 10 alleles using core genome multi-locus sequence typing (cgMLST). Additionally, we have provided a detailed description of this method and its application in our study in the Methods section.

L156, *E. coli* in ital font

Response: Corrected as suggested.

L202, "dependable antibiotic susceptibility outcomes" is unclear.

Response: We apologize for any confusion caused. To ensure the reliability of our antibiotic susceptibility tests, we employed control strains in each test. These control strains serve as benchmarks, and if their susceptibility results do not match the expected outcomes, it indicates that the results are unreliable. Thus, "dependable antibiotic susceptibility outcomes" refer to the results that have passed this quality control check. To make this clearer in the manuscript, we propose the following revision: "Among the 485 isolates with dependable antibiotic susceptibility outcomes (each test was validated using control strains, and only those passing quality control were considered reliable), MDR (multidrug-resistant, resistance to antibiotics from at least three classes) strains accounted for up to 97% of the isolates (Fig. 2a)."

L248, 253, do you count *bla*_{TEM-1B} as ESBL? it is not normally considered as providing broad spectrum resistance.

Response: We acknowledge that *bla*_{TEM-1B} is not typically considered an ESBL gene. In our manuscript, we did not classify *bla*_{TEM-1B} as an ESBL. Specifically, we reported the prevalence of various β -lactamase genes, noting that *bla*_{TEM-1B} was the most prevalent β -lactamase gene, and separately discussed the presence of ESBL genes. We investigated ESBL genes such as *bla*_{CTX-M-14}, *bla*_{CTX-M-55}, *bla*_{CMY-2}, and *bla*_{CTX-M-65}. Detailed results regarding ESBL genes were provided in Supplementary Data 6.

L268, "exhibited the highest resistance" compared to other STs?

Response: Yes, our original sentence intended to convey that, among the four prevalent STs, ST410 exhibited the highest resistance to three fluoroquinolones. To ensure this point is clear, we propose the following revision for added emphasis: "Here, we further combined the comparative results of antibiotic resistance phenotypes among the four prevalent STs, especially noting that strains belonging to ST410 exhibited the highest

resistance to three fluoroquinolones compared to the other STs."

L285-286, unclear how 340 combinations can be present in a single genome.

Response: We understand that the original wording may have caused some confusion. The phrase "340 combinations exclusively present in a single genome" refers to the fact that among the 492 genomes we studied, there were 340 unique combinations of ARGs, each found in only one genome. This means that each of these 340 combinations was distinct and appeared uniquely in a single genome, not that a single genome contained 340 different combinations. To improve clarity, we propose the following revision: "More than 96% of the combinations were contained by only one or two genomes, with 340 genomes each harboring a unique set of ARGs. "

L290, "fighting"? "putative MDR isolates"?

Response: We understand that the term "fighting" may not be appropriate in this context. To improve clarity, we propose the following revision: "When drug classes targeted by these ARGs were predicted, "

Regarding "putative MDR isolates" this terminology is appropriate because here we inferred MDR were based on genetic data predictions rather than antibiotic susceptibility tests.

L302, maybe plural since they are classes and do not include single antibiotics

Response: Corrected as suggested.

L344, located instead of situated.

Response: Corrected as suggested.

L563-565 unclear if you tested for combinations *sfa/foc*, *afa/dra* or these are alternative gene designations.

Response: We apologize for any confusion caused. To clarify, *sfa* and *foc*, as well as *afa* and *dra*, represent alternative gene designations rather than combinations. To improve clarity, we propose the following revision: "*sfa* or *foc* (S or F1C fimbriae), *afa* or *dra* (Dr-antigen-binding adhesins) "

L577, relied

Response: Corrected as suggested.

L579, belonged

Response: Corrected as suggested.

L586, ital font for E. coli

Response: Corrected as suggested.

Reviewer #3 (Remarks to the Author):

In this second version, the authors have considered all the reviewers' suggestions to complete the required information and to perform a deeper analysis of their results. From my point of view, this new version has improved in terms of reinforcing the results and conclusions obtained by their research.

All the issues posed have been satisfactorily addressed, so I recommend the publication of this work in its current state.